# Towards Dynamic Graph Neural Networks with Provably High-Order Expressive Power

## Abstract

Dynamic Graph Neural Networks (DyGNNs) have garnered increasing research attention for learning representations on evolving graphs. Despite their effectiveness, the limited expressive power of existing DyGNNs hinders them from capturing important evolving patterns of dynamic graphs. Although some works attempt to enhance expressive capability with heuristic features, there remains a lack of DyGNN frameworks with provable and quantifiable high-order expressive power. To address this research gap, we firstly propose the $k$-dimensional Dynamic WL tests ($k$-DWL) as the referencing algorithms to quantify the expressive power of DyGNNs. We demonstrate that the expressive power of existing DyGNNs is upper bounded by the 1-DWL test. To enhance the expressive power, we propose **D**ynamic **G**raph Neural **N**etwork with **H**igh-**o**rder **e**xpressive **p**ower (**HopeDGN**), which updates the representation of central node pair by aggregating the interaction history with neighboring node pairs. Our theoretical results demonstrate that HopeDGN can achieve expressive power equivalent to the 2-DWL test. We then present a Transformer-based implementation for the local variant of HopeDGN. Experimental results show that HopeDGN achieved performance improvements of up to 3.12%, demonstrating the effectiveness of HopeDGN.

## 1 Introduction

Graph Neural Networks (GNNs) have emerged as dominant tools for learning low-dimensional representations of graph-structured data (Kipf & Welling, 2017; Veličković et al., 2017; Hamilton et al., 2017; Gasteiger et al., 2018). However, many real-world graphs exhibit dynamic properties with continuously evolving topological structures. Due to this prevalence, an increasing number of research has focused on learning effective representations of dynamic graphs using Dynamic Graph Neural Networks (DyGNNs). Most DyGNNs employ a message-passing framework, where historically interacted nodes are aggregated using techniques such as sum-pooling (Wen & Fang, 2022), local self-attention (Xu et al., 2020; Fan et al., 2021), and Transformers (Yu et al., 2023). DyGNNs have been successfully applied to various tasks such as financial fraud detection (Huang et al., 2022), traffic prediction (Han et al., 2021), and sequential recommendation (Kumar et al., 2019).

One crucial requirement for designing (dynamic) GNNs is sufficient expressive power; that is, the (dynamic) GNNs should be capable of distinguishing non-isomorphic (dynamic) graphs. Xu et al. (2019) and Morris et al. (2019) underscored that the expressive power of message-passing-based GNNs is bounded by the 1-Weisfeiler-Lehman (WL) test, which prompts extensive studies on GNNs with expressive power beyond 1-WL test (Maron et al., 2019a; Zhang et al., 2024). However, these investigations have predominantly focused on static graphs. As we discuss in Section 4.1, existing DyGNNs remain facing limitations in expressive power when applied to dynamic graphs. Consequently, existing DyGNNs fail to detect some evolving substructures such as triangle structures (an illustrative example is provided in Figure 1), which are important for capturing the evolution patterns of dynamic graphs (Paranjape et al., 2017; Zhou et al., 2018; Zitnik et al., 2019). Few works have targeted at designing DyGNNs with stronger expressive power. Souza et al. (2022) proposed relative positional features to enhance the expressive power of DyGNNs. However, from a theoretical perspective, it remains unclear how the relative positional features quantitatively affect DyGNNs' expressive power. *To summarize, how to design DyGNNs with provably and quantitatively high-order expressive power remains unexplored.*

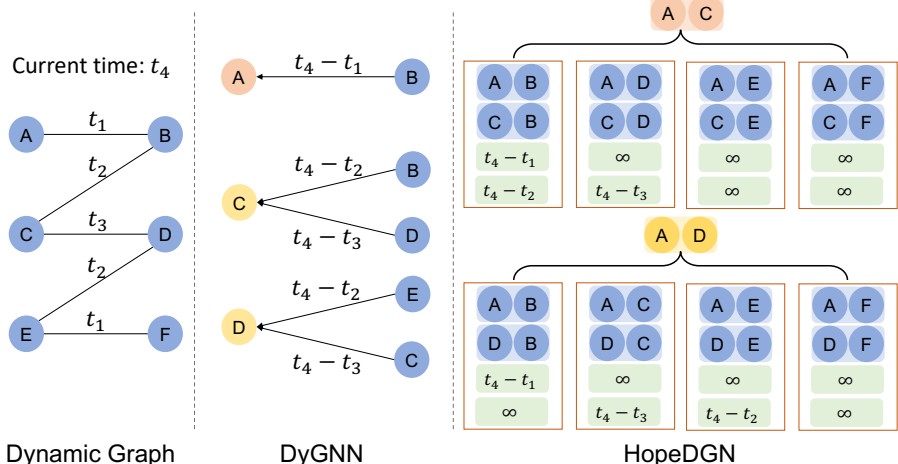

Figure 1: An example of limited expressive power of DyGNNs. Suppose the model is distinguishing node pairs $(A, C)$ and $(A, D)$ at time $t_4$. Because $A$ and $C$ have historical interaction with $B$ while $A$ and $D$ do not have common historical interacted nodes, $(A, C)$ and $(A, D)$ are not isomorphic at time $t_4$. Since nodes $C$ and $D$ are isomorphic on the historical interaction graph before $t_4$, DyGNNs will output the same embeddings for $(C, t_4)$ and $(D, t_4)$. Thus, DyGNNs fails to distinguish $(A, C, t_4)$ and $(A, D, t_4)$. Conversely, HopeDGN will notice that node $B$ interacts with $A$ and $C$ at $t_1$ and $t_2$ respectively, thus being capable of distinguish these node pairs.

To address this research gap, we begin by presenting a theoretical framework to quantify the expressive power of existing DyGNNs. Specifically, we extend the Weisfeiler-Lehman (WL) hierarchy tests and propose the $k$-dimensional Dynamic WL (DWL) tests ($k \geq 1$) as the referencing algorithms to check the isomorphism on dynamic graphs. We demonstrate that the expressive power of existing DyGNNs is upper bounded by the proposed 1-DWL test. To enhance the expressive power of existing DyGNNs, we propose the Multi-Interacted Time Encoding (MITE), which encodes the bi-interaction history of target node pairs with other nodes, thereby capturing the indirect dependencies between target node pairs. MITE is a plug-and-play module that can be seamlessly integrated into a wide range of models. Equipped with MITE, we introduce the **D**ynamic **G**raph Neural **N**etwork with **H**igh-**o**rder **e**xpressive **p**ower (**HopeDGN**), which updates the representations of target node pairs by aggregating their neighboring node pairs as well as the multi-interaction history. Our theoretical results demonstrate that HopeDGN can achieve expressive power equivalent to the 2-DWL test with injective aggregation and updating functions. We further present a Transformer-based implementation of the local version of the proposed HopeDGN. Experimental results demonstrate that the proposed HopeDGN achieves superior performance on seven datasets compared to other baselines, underscoring the effectiveness of the proposed HopeDGN. In summary, the main contributions of this work are three-fold:

- We establish a theoretical framework to quantify the expressive power of DyGNNs, and prove that the expressive power of existing DyGNNs is upper bounded by the 1-DWL test.

- We propose HopeDGN which can achieve expressive power equivalent to the 2-DWL test, thus being provably and quantitatively more expressive than existing DyGNNs.

- Extensive experiments on both link prediction and node classification tasks demonstrate the superiority of the proposed HopeDGN over existing models.

## 2 RELATED WORKS

**Dynamic Graphs Neural Networks.** A dynamic graph is a network whose topological structure or node attributes evolve over time. Depending on whether the timestamps are discrete or continuous, dynamic graphs can be categorized into Discrete-Time Dynamic Graphs (DTDGs) and Continuous-Time Dynamic Graphs (CTDGs). Existing studies on DTDGs typically integrate Graph

Neural Networks (GNNs) with sequential models to learn structural representations of graph snapshots and their evolution patterns (Yu et al., 2017; Sankar et al., 2020; You et al., 2022; Zhu et al., 2023; Zhang et al., 2023b). Some approaches also utilize sequential models to update the weights of GNNs (Pareja et al., 2020). Recently, CTDGs have emerged as a more general form of dynamic graphs, garnering increasing attention from the research community. Most existing works on CT-DGs adopt an aggregation-then-update framework (see Section 3) (Wen & Fang, 2022; Souza et al., 2022). Various aggregation techniques have been proposed, including local self-attention (Xu et al., 2020), Transformers (Yu et al., 2023; Wang et al., 2024), and MLP-mixers (Cong et al., 2023). Memory mechanisms have also been employed to retain long-term interaction information (Kumar et al., 2019; Trivedi et al., 2019; Rossi et al., 2020). Additionally, some studies leverage temporal random walks to learn representations (Wang et al., 2021b; Jin et al., 2022). Compared to existing works, the proposed HopeDGN learns representations of node pairs rather than individual nodes. More importantly, the proposed HopeDGN achieves the equivalent expressive power of the 2-DWL test, which is significantly more powerful than existing DyGNNs.

**Expressive power of GNNs.** The expressive power of Graph Neural Networks (GNNs) is measured by their ability to distinguish non-isomorphic graphs. Since the seminal works of Xu et al. (2019) and Morris et al. (2019) demonstrated that the expressive power of message-passing based GNNs is upper-bounded by the 1-WL test, extensive efforts have been made to enhance the expressive power of GNNs. Some methods proposed high-order GNNs that mimic the procedure of higher-order WL tests (Maron et al., 2018; 2019b; Azizian & Lelarge, 2020; Geerts & Reutter, 2022). Other methods aggregated the learned node representations on pre-generated subgraphs (Cotta et al., 2021; Zhao et al., 2021; Bevilacqua et al., 2021). Furthermore, some works incorporated substructure information into the learning of node representations (Chen et al., 2020; Bouritsas et al., 2022; Horn et al., 2021). Zhang et al. (2023a) also proposed evaluating the expressive power of GNNs via graph biconnectivity. While the expressive power of static GNNs has been extensively studied, few works have investigated the expressive power of Dynamic GNNs (DyGNNs). Souza et al. (2022) proposed a DyGNN with an expressive power equivalent to the 1-Temporal WL test, further enhanced by relative position features. Gao & Ribeiro (2022) studied the equivalent expressive power of two types of dynamic graphs, namely time-then-graph and time-and-graph. Despite these efforts, DyGNNs with quantifiable high-order expressiveness are still lacking.

## 3  PRELIMINARIES

**Graph Isomorphism.** A *graph* is defined as $\mathcal{G} = \{\mathcal{V}, \mathcal{E}\}$ where $\mathcal{V} = \{1, 2, ..., N\}$ is the node set and $\mathcal{E} = \{(u, v) \subseteq \mathcal{V} \times \mathcal{V}\}$ is the edge set. The $k$-node tuple is defined as $\boldsymbol{s} = (v_1, ..., v_k)$ with $v_i \in \mathcal{V}$ and all $k$-node tuples constitutes the set $[\mathcal{V}]^k$. The neighbor set of node $u$ is defined as $\mathcal{N}(u) = \{v | (u, v) \in \mathcal{E} \vee (v, u) \in \mathcal{E}\}$. Two graphs $\mathcal{G} = \{\mathcal{V}, \mathcal{E}\}$ and $\mathcal{G}' = \{\mathcal{V}', \mathcal{E}'\}$ are said *isomorphic* if there exists a bijective mapping $\varphi : \mathcal{V} \to \mathcal{V}'$ such that $(u, v) \in \mathcal{E}$ if and only if $(\varphi(u), \varphi(v)) \in \mathcal{E}'$, denoted as $\mathcal{G} \cong \mathcal{G}'$. If $\mathcal{G}$ and $\mathcal{G}'$ are the same graphs, we call $\varphi$ an *automorphism*. Given two $k$-node tuple $\boldsymbol{s}$ and $\boldsymbol{s}'$, we say $\boldsymbol{s}$ and $\boldsymbol{s}'$ are *isomorphic* if there exists a graph isomorphic mapping $\varphi : \mathcal{V} \to \mathcal{V}'$ such that $u \in S$ if and only if $\varphi(u) \in S'$. A *labeling* of $\mathcal{G}$ is a function that maps a $k$-node tuple $\boldsymbol{s}$ to a label: $l : [\mathcal{V}]^k \to \mathbb{N}$.

**Dynamic Graph.** Unless otherwise specified, we use Dynamic Graph to denote Continuous-Time Dynamic Graph in the following sections. A *Dynamic Graph* is defined as $\mathcal{DG} = (\mathcal{V}, \mathcal{E})$, where $\mathcal{V} = \{1, 2, ..., N\}$ is the node set and $\mathcal{E} = \{(u_1, v_1, t_1), (u_2, v_2, t_2), ...\}$ with $t_i \leq t_{i+1}$ is a sequence of node interactions labeled with timestamps. $(u_i, v_i, t_i)$ represents that node $u_i$ and node $v_i$ have an interaction event at time $t_i$. The node feature matrix of $\mathcal{DG}$ is denoted $\boldsymbol{X} \in \mathbb{R}^{|\mathcal{V}| \times d_N}$, and the edge feature matrix is denoted as $\boldsymbol{E} \in \mathbb{R}^{d_E}$. For datasets without predefined node (edge) features, the node (edge) features are set as zero vectors. Note that the same node pair may interact multiple times in the dynamic graph. Given the historical interactions before time $t$, we aim to learn the temporal embeddings of each $k$-node tuple $\boldsymbol{s} \in [\mathcal{V}]^k$ at time $t$ with a mapping function $f : [\mathcal{V}]^k \to \mathbb{R}^d$. $k = 1$ and $k = 2$ correspond to the temporal node embeddings and edge embeddings, respectively. The learned temporal embeddings can be leveraged for downstream tasks such as link prediction and node classification.

**Dynamic Graph Neural Networks (DyGNNs).** The workflows of the DyGNN consist of two modules: AGG and UPDATE. The AGG module aggregates the messages of historical neighbors. The aggregation is then passed to the UPDATE module to update the embedding of the root node. Specifically, the *historical neighbor* of node $u$ at time $t$ is defined as $\mathcal{N}(u,t) = \{(w,t')|t' < t, (u,w,t') \in \mathcal{E} \vee (w,u,t') \in \mathcal{E}\}$. The 0-th layer embedding of node $u$ is the node feature $\boldsymbol{h}_t^{(0)}(u) = \boldsymbol{X}_u$. The $l$-th layer ($l > 0$) embedding is computed as:

$$\tilde{\boldsymbol{h}}_t^{(l)}(u) = \text{AGG}(\{\!\{([\boldsymbol{h}_{t'}^{(l-1)}(w)||\sigma(t-t')]|(w,t') \in \mathcal{N}(u,t)\}\!\}),$$
$$\boldsymbol{h}_t^{(l)}(u) = \text{UPDATE}(\boldsymbol{h}_t^{(l-1)}(u), \tilde{\boldsymbol{h}}_t^{(l)}(u)) \tag{1}$$

where $\sigma : \mathbb{R}^+ \to \mathbb{R}^{d_T}$ projects the time interval to a vector and $||$ denotes the concatenation. $\{\!\{\cdot\}\!\}$ denotes the multiset. AGG can be implemented as local self-attention (Vaswani et al., 2017; Xu et al., 2020), MLP-Mixer (Tolstikhin et al., 2021; Cong et al., 2023), etc. For link prediction task, to predict the existence of interaction $(u,v)$ at time $t$, the temporal embeddings $\boldsymbol{h}_t(u)$ and $\boldsymbol{h}_t(v)$ are merged to generate the probability. Some methods (Kumar et al., 2019; Rossi et al., 2020) also leverage memory mechanisms to record the long-term historical interactions of each node. Specifically, the memory state of node $u$ at $t = 0$ is initialized as $\boldsymbol{s}_0(u) = \boldsymbol{X}_u$. When an interaction associated with $u$ happens, say $(u,v,t)$, $\boldsymbol{s}_u$ is updated as:

$$\boldsymbol{m}_t(u) = \text{MSG}(\boldsymbol{s}_{t^-}(u), \boldsymbol{s}_{t^-}(v), \Delta t, \boldsymbol{e}_{ij}(t))$$
$$\boldsymbol{s}_t(u) = \text{MEMUPD}(\boldsymbol{s}_{t^-}(u), \boldsymbol{m}_t(u)) \tag{2}$$

where MSG is a message function implemented as Multi-Layer Perception (MLP) or identity, and $\Delta t$ is the time interval since last update. MEMUPD is a memory update function usually implemented as a recurrent neural network such as GRU (Cho et al., 2014). With the memory state, the 0-th layer node embedding is modified as $\boldsymbol{h}_t^{(0)}(u) = \boldsymbol{s}_t(u)$.

## 4 METHODS

In this section, we propose the $k$-Dynamic WL (DWL) test based on the isomorphism on dynamic graphs, and prove that the expressive power of DyGNNs is upper bounded by 1-DWL test (Sec. 4.1). To enhance the expressive power, we propose MITE, which allows DyGNNs to capture the temporal dependency between node pairs (Sec. 4.2). Equipped with MITE, we propose HopeDGN, which is as powerful as 2-DWL test (Sec. 4.3). Finally, we present a Transformer-based implementation of the local HopeDGN(Sec. 4.4). Proofs for all propositions are provided in Appendix B.

### 4.1 LIMITED EXPRESSIVE POWER OF DYGNN

In this section, we study the expressive power of DyGNNs, which are characterized by their capabilities to distinguish non-isomorphic dynamic graphs. In contrast to static graphs, the isomorphism of two dynamic graphs requires that the complete interaction time sequences of corresponding nodes are identical. However, most existing DyGNNs process mini-batches of interactions in chronological order, making it challenging to capture the global evolving structure of dynamic graphs. To this end, we propose *Dynamic Adjacency Tensor*, which represents the interactions within the dynamic graph as a timestamp-labeled multigraph.

**Dynamic Adjacency Tensor.** Let $\mathcal{DG} = \{\mathcal{V}, \mathcal{G}\}$ be a dynamic graph and $T$ be the maximum interaction counts among all node pairs . The *Dynamic Adjacency Tensor (DAT)* of $\mathcal{DG}$ is defined as a tensor $\boldsymbol{A} \in \mathbb{R}^{|\mathcal{V}| \times |\mathcal{V}| \times T}$, where $\boldsymbol{A}_{u,v,:} = [t_1, t_2, ..., t_{q(u,v)}, \infty, ..., \infty]$ with $t_i \le t_{i+1}$ recording $(u,v)$'s interaction timestamp sequence $\{t_1, t_2, ..., t_{q(u,v)}\}$. $q(u,v)$ is the interaction count of the node pair $(u,v)$. $\infty$ is padded if $q(u,v) < T$. In addition, given the current time $t$, to depict the interaction timestamps before $t$, we define the *Historical DAT (HDAT)* as:

$$\boldsymbol{A}_{i,j,k}^{<t} = \begin{cases} \boldsymbol{A}_{i,j,k} & \text{if } \boldsymbol{A}_{i,j,k} < t \\ \infty & \text{else} \end{cases} \tag{3}$$

**Isomorphism on Dynamic Graphs.** With DAT, we are now ready to define the isomorphism on dynamic graphs. Let $\mathcal{DG} = \{\mathcal{V}, \mathcal{E}\}$ and $\mathcal{DG}' = \{\mathcal{V}', \mathcal{E}'\}$ be two dynamic graphs, and $\boldsymbol{A}$ and $\boldsymbol{A}'$ be

their corresponding DATs. We say $\mathcal{DG}$ and $\mathcal{DG}'$ are *isomorphic* if there exists a bijective mapping $\varphi$: $\mathcal{V} \to \mathcal{V}'$ such that $\mathbf{A}_{i,j,:} = \mathbf{A}'_{\varphi(i),\varphi(j),:}$ for all $(i,j) \in \mathcal{V} \times \mathcal{V}$, denoted as $\mathcal{DG} \cong \mathcal{DG}'$. If $\mathcal{DG} = \mathcal{DG}'$, we call $\varphi$ an *automorphism*. Considering the HDAT, we say $\mathcal{DG}$ and $\mathcal{DG}'$ are **isomorphic until $t$** if there exists bijective mapping $\varphi\colon \mathcal{V} \to \mathcal{V}'$ such that $\mathbf{A}^{<t}_{i,j,:} = \mathbf{A}'^{<t}_{\varphi(i),\varphi(j),:}$ for all $(i,j) \in \mathcal{V} \times \mathcal{V}$, denoted as $(\mathcal{DG}, t) \cong (\mathcal{DG}', t)$. Additionally, we say two $k$-node tuples $\boldsymbol{s} \in [\mathcal{V}]^k$ and $\boldsymbol{s}' \in [\mathcal{V}']^k$ are isomorphic if there exists a bijection $\varphi$ from $\boldsymbol{s}$ to $\boldsymbol{s}'$ and $\varphi$ is an isomorphism from $\mathcal{DG}$ to $\mathcal{DG}'$.

It is challenging to quantify the number of non-isomorphic dynamic graphs that an algorithm can distinguish. The Weisfeiler-Lehman (WL) test (Leman & Weisfeiler, 1968) is a classical algorithm for determining graph isomorphism and is widely used to quantify the expressive power of Graph Neural Networks (GNNs) (Xu et al., 2019; Morris et al., 2019). To quantify the expressive power of DyGNNs, we extend WL tests to dynamic graphs and propose Dynamic WL tests.

**Dynamic WL (DWL) tests.** To compute the color of the center node at a specific time, the 1-DWL test aggregates the color and complete interaction history of its neighbors, then hashes them into a unique node color. Specifically, given a dynamic graph $\mathcal{DG} = \{\mathcal{V}, \mathcal{E}\}$ and a node labeling function $l : \mathcal{V} \to \mathbb{N}$ at timestamp $t$, the 1-DWL test initializes the node color at $t$ as $c_t^{(0)}(u) = l(u)$. Then, at $j$-th iteration ($j > 0$), the node color is refined as:

$$c_t^{(j)}(u) = \text{HASH}(c_t^{(j-1)}(u), \{\!\!\{(c_t^{(j-1)}(w), \mathbf{A}^{<t}_{u,v,:})|(v,\cdot) \in \mathcal{N}(u,t)\}\!\!\}) \tag{4}$$

where HASH is a hashing function. To test whether two graphs $\mathcal{DG}$ and $\mathcal{DG}'$ are isomorphic until $t$, we run 1-DWL test on both graphs in parallel. If the multisets of node colors in two graphs are not equal at any iteration, the 1-DWL test concludes that $\mathcal{G}$ and $\mathcal{G}'$ are not isomorphic until $t$. In addition, the $k$-DWL ($k \geq 2$) tests process as follows. Let $\boldsymbol{s} = (v_1, ..., v_k)$ be a $k$-node tuple and $l$ be a node tuple labeling function, the $k$-DWL test initializes the node color of each $k$-node tuple as $c_t^{(0)}(\boldsymbol{s}) = l(\boldsymbol{s})$. Then, at $j$-th iteration ($j > 0$), the color of node tuple is refined as:

$$c_t^{(j)}(\boldsymbol{s}) = \text{HASH}\big(c_t^{(j-1)}(\boldsymbol{s}), \{\!\!\{\boldsymbol{\phi}_t^{(j-1)}(\boldsymbol{s}, w)|w \in \mathcal{V}\}\!\!\}\big)$$
$$\boldsymbol{\phi}_t^{(j-1)}(\boldsymbol{s}, w) = \big(c_t^{(j-1)}(\boldsymbol{r}_1(\boldsymbol{s}, w)), ..., c_t^{(j-1)}(\boldsymbol{r}_k(\boldsymbol{s}, w)), \mathbf{A}^{<t}_{w,v_1,:}, ..., \mathbf{A}^{<t}_{w,v_k,:}\big) \tag{5}$$

where $\boldsymbol{r}_i(\boldsymbol{s}, w) = (v_1, ..., v_{i-1}, w, v_{i+1}, ..., v_k)$. The following procedures work analogously to 1-DWL. Here the "neighboring node tuple" of $\boldsymbol{s}$ is obtained by replacing each element in $\boldsymbol{s}$ with other nodes. Intuitively, $k$-DWL test refines the color of the central $k$-node tuple at time $t$ by aggregating the colors and complete interaction history of neighboring node tuples. Note that the proposed $k$-DWL test has a similar procedure as the Folklore variant of $k$-WL test (Cai et al., 1992) in static graphs, which groups and hashes the node tuple with the same replacing nodes. The following proposition states that $(k+1)$-DML test is at least as powerful as $k$-DWL test in distinguishing non-isomorphic dynamic graphs ($k \geq 1$), which demonstrates that the proposed $k$-DWL tests provide a valid hierarchical framework for checking the isomorphism of dynamic graphs.

**Proposition 1.** *Let $\mathcal{DG} = \{\mathcal{V}, \mathcal{E}\}$ and $\mathcal{DG}' = \{\mathcal{V}', \mathcal{E}'\}$ be two dynamic graphs. Suppose the initial labeling function of $k$-DWL test be constant. Then, for all $k \geq 1$, if $k$-DWL test decides $\mathcal{DG}$ and $\mathcal{DG}'$ are non-isomorphic, then $(k+1)$-DWL test also decides $\mathcal{DG}$ and $\mathcal{DG}'$ are non-isomorphic.*

Next, we show the expressive power of existing DyGNNs is strictly bounded by 1-DWL test. Specifically, at any iterations of 1-DWL test and DyGNNs, if 1-DWL assigns the same colors for nodes $u$ and $v$ at time $t$, then DyGNN will also output the same temporal embeddings of $u$ and $v$ at time $t$.

**Proposition 2.** *Let $\mathcal{DG} = \{\mathcal{V}, \mathcal{E}\}$ and $\mathcal{DG}' = \{\mathcal{V}', \mathcal{E}'\}$ be two dynamic graphs, and $\boldsymbol{X}$ and $\boldsymbol{X}'$ be their corresponding node features. Given a node labeling function $l : \mathcal{V} \to \mathbb{N}$ satisfying $l(u) = l(v)$ if and only if $\boldsymbol{X}_u = \boldsymbol{X}'_v$ for any $u \in \mathcal{V}$ and $v \in \mathcal{V}'$. Let $c_t^{(j)}$ denotes the color at time $t$ obtained by 1-DWL test initialized with label function $l$ in the $j$-th iteration, and $\boldsymbol{h}_t^{(j)}$ be the temporal node embeddings outputted by the DyGNN. Then for all $j \geq 0$, $c_t^{(j)}(u) = c_t^{(j)}(v) \implies \boldsymbol{h}_t^{(j)}(u) = \boldsymbol{h}_t^{(j)}(v)$.*

Souza et al. (2022) proves that adding a memory mechanism will not change the expressive power of DyGNNs. Therefore, the expressive power of DyGNNs can be fully characterized by the 1-DWL test. Although the 1-DWL test is effective in detecting two non-isomorphic nodes in dynamic graphs, it often fails to detect two non-isomorphic multi-node tuples. The reason is that the 1-DWL test

independently aggregates the historical neighbors of each node, but ignores the evolving dependency between multiple nodes such as common historical neighbors (see the example in Fig. 1). These indirect dependencies are important for multi-node level tasks such as future link prediction.

## 4.2 Multi-Interacted Time Encoding

As stated in the previous section, 1-DWL and DyGNNs cannot capture the dependencies between multiple nodes. To address this limitation, we propose Multi-Interacted Time Encoding (MITE). Intuitively, MITE encodes the complete bi-interaction history of target node pairs with other nodes in the dynamic graph, thereby capturing dependency information such as common neighbors. Unlike static graphs, dynamic graphs may have multiple interactions between two nodes at different timestamps. Encoding the interaction time series provides valuable information, such as interaction frequency and the time interval since the last interaction, which aids in learning better representations. Specifically, Let $\mathcal{DG} = \{\mathcal{V}, \mathcal{E}\}$ be a dynamic graph and its DAT is denoted as $\mathbf{A}$. At time $t$, the *Time Interval Tensor (TIT)* $\mathbf{B}^t \in \mathbb{R}^{|\mathcal{V}| \times |\mathcal{V}| \times T}$ is computed as:

$$\mathbf{B}^t_{i,j,k} = \begin{cases} t - \mathbf{A}^{<t}_{i,j,k} & \text{if } \mathbf{A}^{<t}_{i,j,k} < t \\ \infty & \text{else} \end{cases} \tag{6}$$

Given the target node pair $\boldsymbol{s} = (u, v)$ at time $t$, its MITE with respect to a node $w \in \mathcal{V}$ is defined as:

$$\boldsymbol{X}^t_{M,w} = f([\mathbf{B}^t_{w,u,:} || \mathbf{B}^t_{w,v,:}]) \in \mathbb{R}^{d_B} \tag{7}$$

where $f(\cdot)$ is implemented as a two-layer MLP in our work. For implementations, a normalization operator such as logarithm is applied to $\mathbf{B}$ due to its possible large variance. In addition, as the maximum interaction count $T$ may be very large, we preserve the last $K(K < T)$ non-infinite timestamps of $\mathbf{B}_{w,\cdot,:}$. $\mathbf{B}_{w,\cdot,:}$ is padded if the number of non-infinite timestamps is less than $K$.

The proposed MITE can be integrated with existing DyGNNs by incorporating it with node features. As such, the DyGNN model will capture the dependency information of node pairs, which enhance the expressive power of original DyGNNs. The following proposition shows a case of non-isomorphic node pairs that DyGNNs with MITE can distinguish while vanilla DyGNNs cannot.

**Proposition 3.** *There exists two dynamic graphs $\mathcal{DG} = \{\mathcal{V}, \mathcal{E}\}$ and $\mathcal{DG}' = \{\mathcal{V}', \mathcal{E}'\}$ which have non-isomorphic node pairs $\boldsymbol{s} \in [\mathcal{V}]^2$ and $\boldsymbol{s}' \in [\mathcal{V}']^2$ until some time $t$ that DyGNN with MITE can distinguish while vanilla DyGNN cannot.*

**Connections with Neighbor Co-Occurrence Encoding (Yu et al., 2023).** Yu et al. (2023) proposed the Neighbor Co-Occurrence Encoding (NCOE) which encodes the interaction count of the target node pairs to other nodes. For example, suppose the historical interaction sequences of nodes $u$ and $v$ are $\{a, b, a\}$ and $\{b, b, a, c\}$, respectively, then the NCOEs of $a$, $b$, $c$ are $[2, 1]$, $[1, 2]$, $[0, 1]$, respectively. Note that MITE degenerates to NCOE by setting $f$ in Eq. (7) to output the number of non-infinity elements. Compared to NCOE, MITE additionally captures the timestamps information of bi-interaction, which contains richer semantic information.

## 4.3 Dynamic Graph Neural Network with High-order expressive power

Equipped with MITE, in this section, we propose the **D**ynamic **G**raph Neural **N**etwork with **H**igh-**o**rder **e**xpressive **p**ower (**HopeDGN**), which works analogously to the 2-DWL test. HopeDGN update the temporal embedding of a central *node pair* by aggregating its neighboring node pairs as well as the their interaction history with central node pair. Specifically, given the dynamic graph $\mathcal{DG} = \{\mathcal{V}, \mathcal{E}\}$ and the node feature $\boldsymbol{X}$. The TIT at time $t$ is denoted as $\mathbf{B}^t$. The 0-th layer temporal embedding of the node pair $\boldsymbol{s} = (u, v)$ at time $t$ is $\boldsymbol{h}^{(0)}_t(\boldsymbol{s}) = [\boldsymbol{X}_u || \boldsymbol{X}_v]$. Then, the $l$-th layer ($l > 0$) embedding of $\boldsymbol{s}$ at time $t$ is computed as:

$$\begin{aligned} \boldsymbol{h}^{(l)}_t(\boldsymbol{s}) &= \text{UPDATE}\big(\boldsymbol{h}^{(l-1)}_t(\boldsymbol{s}), \tilde{\boldsymbol{h}}^{(l)}_t(\boldsymbol{s})\big) \\ \tilde{\boldsymbol{h}}^{(l)}_t(\boldsymbol{s}) &= \text{AGG}\Big(\{\!\!\{\ \boldsymbol{\psi}_t(\boldsymbol{s}, w) \mid w \in \mathcal{V}\ \}\!\!\}\Big) \\ \boldsymbol{\psi}_t(\boldsymbol{s}, w) &= \big[f_1\big([\boldsymbol{h}^{(l-1)}_t((u, w)) || \boldsymbol{h}^{(l-1)}_t((v, w))]\big) || \underbrace{f_2\big([\mathbf{B}^t_{u,w,:} || \mathbf{B}^t_{v,w,:}]\big)}_{\text{MITE of } w}\big] \end{aligned} \tag{8}$$

where $f_1$ and $f_2$ are projecting functions, which are implemented as MLPs in our work. Here the replacing node $w$ can be chosen from entire node set $\mathcal{V}$, thus we call this formulation as Global HopeDGN. However, the number of nodes may be enormous on large-scale dynamic graphs, and the computation cost of Eq. (8) may be expensive. Therefore, we propose a local version of HopeDGN, which only takes the historical neighbors of $u$ and $v$ as replacing nodes:

$$
\begin{aligned}
\boldsymbol{h}_t^{(l)}(\boldsymbol{s}) &= \text{UPDATE}\big(\boldsymbol{h}_t^{(l-1)}(\boldsymbol{s}), \tilde{\boldsymbol{h}}_t^{(l)}(\boldsymbol{s})\big) \\
\tilde{\boldsymbol{h}}_t^{(l)}(\boldsymbol{s}) &= \text{AGG}\Big(\{\!\!\{ \ \boldsymbol{\psi}_t(\boldsymbol{s}, w) \mid (w, \cdot) \in \mathcal{N}(u, t) \cup \mathcal{N}(v, t) \ \}\!\!\}\Big)
\end{aligned}
\tag{9}
$$

Similar to Proposition 2, we will now show that the expressive power of HopeDGN is upper bounded by 2-DWL test, i.e., any non-isomorphic node pairs that can be distinguished by HopeDGN will also be distinguished by 2-DWL test.

**Proposition 4.** *Let $\mathcal{DG} = \{\mathcal{V}, \mathcal{E}\}$ and $\mathcal{DG}' = \{\mathcal{V}', \mathcal{E}'\}$ be two dynamic graphs, and $\boldsymbol{X}$ and $\boldsymbol{X}'$ be their corresponding node features. Given a node labeling function $l : [\mathcal{V}]^2 \to \mathbb{N}$ satisfying $l((u, v)) = l((u', v'))$ if and only if $[\boldsymbol{X}_u || \boldsymbol{X}_v] = [\boldsymbol{X}'_{u'} || \boldsymbol{X}'_{v'}]$ for all $(u, v) \in [\mathcal{V}]^2$ and $(u', v') \in [\mathcal{V}']^2$. Let $c_t^{(j)}$ denotes the color at time $t$ obtained by 2-DWL test , initialized with label function $l$ in the $j$-th iteration, and $\boldsymbol{h}_t^{(j)}$ be the temporal node embeddings output by the Global HopeDGN. Then for all $j \geq 0$, $c_t^{(j)}\big((u, v)\big) = c_t^{(j)}\big((u', v')\big) \Longrightarrow \boldsymbol{h}_t^{(j)}\big((u, v)\big) = \boldsymbol{h}_t^{(j)}\big((u', v')\big)$.*

Additionally, we will prove that if the UPDATE , AGG, $f_1$ and $f_2$ in Eq. (8) meet the injective requirement, the HopeDGN is as powerful as the 2-DWL test, as shown in following proposition.

**Proposition 5.** *Let $\mathcal{M} : [\mathcal{V}]^2 \to \mathbb{R}^d$ be a Global HopeDGN. Suppose the 2-DWL test is initialized with a node labeling function $l : [\mathcal{V}]^2 \to \mathbb{N}$ satisfying $l((u, v)) = l((u', v'))$ if and only if $[\boldsymbol{X}_u || \boldsymbol{X}_v] = [\boldsymbol{X}'_{u'} || \boldsymbol{X}'_{v'}]$ for all $(u, v) \in [\mathcal{V}]^2$ and $(u', v') \in [\mathcal{V}']^2$. If the AGG, UPDATE, $f_1$ and $f_2$ of $\mathcal{M}$ are injective, then at any time $t$, if 2-DWL test assigns different colors to two node pairs, $\mathcal{M}$ will also output different temporal embeddings of these two node pairs.*

The injectiveness of each function in Global HopeDGN can be approximated with MLP or other neural networks due to the universal approximation theorem Hochreiter & Schmidhuber (1997).

## 4.4 IMPLEMENTATION DETAILS

In this section, we present the implementation details of local HopeDGN. Considering that some interactions may happen long time ago, we leverage Transformer (Vaswani et al., 2017) as the backbone due to its capability of modeling long-term dependency.

**Neighborhood Encodings.** Let $\mathcal{DG} = \{\mathcal{V}, \mathcal{E}\}$ be a dynamic graph, and its node feature and edge feature are denoted as $\boldsymbol{X} \in \mathbb{R}^{|\mathcal{V}| \times d_N}$ and $\boldsymbol{E} \in \mathbb{R}^{|\mathcal{E}| \times d_E}$, respectively. Based on Eq. (8), given the target node pair $\boldsymbol{s} = (u, v)$ at time $t$, we need to aggregate the joint neighborhood of $u$ and $v$, denoted as $\mathcal{N}(u, t)$ and $\mathcal{N}(v, t)$, respectively. Here we only learn from the one-hop joint neighborhood for efficiency of computation. For each $(w, t') \in \mathcal{N}(u, t) \cup \mathcal{N}(v, t)$, the combined node encodings of $w$ is represented as $\boldsymbol{X}_{C,w} = [\boldsymbol{X}_w || \boldsymbol{X}_u || \boldsymbol{X}_v] \in \mathbb{R}^{d_C}$. The edge encodings of $(w, t')$ are retrieved from $\boldsymbol{E}$, denoted as $\boldsymbol{X}_{E,w} \in \mathbb{R}^{d_E}$. Following Xu et al. (2020), the time encoding of $w$ is learned by applying random Fourier feature on time interval $\Delta t = t - t'$, computed as $\boldsymbol{X}_{T,w} = \sqrt{2/d_T}[\cos(w_1 \Delta t), \sin(w_1 \Delta t), ..., \cos(w_{d_{T/2}} \Delta t), \sin(w_{d_{T/2}} \Delta t)] \in \mathbb{R}^{d_T}$. The MITE of $w$ with respect to $(u, v)$ is denoted as $X_{B,w} \in \mathbb{R}^{d_B}$ (Section 4.2). The corresponding encodings of the complete joint neighborhood are denoted as $\boldsymbol{X}_C \in \mathbb{R}^{S \times d_C}$, $\boldsymbol{X}_E \in \mathbb{R}^{S \times d_E}$, $\boldsymbol{X}_T \in \mathbb{R}^{S \times d_T}$, $\boldsymbol{X}_M \in \mathbb{R}^{S \times d_M}$ with $S = |\mathcal{N}(u, t) \cup \mathcal{N}(v, t)|$.

**Patching Technique.** Since the length of joint neighborhood may be very large, inspired by Dosovitskiy et al. (2021), we leverage the patching technique to divide the neighborhood sequence into non-overlapping patches. Let $P$ denote the patch size. we take the combined node encoding $\boldsymbol{X}_C$ as an example. $\boldsymbol{X}_C \in \mathbb{R}^{S \times d_C}$ will be reshaped into $\mathbb{R}^{N_p \times (P \cdot d_C)}$ with $N_p = \lceil S/P \rceil$ (neighborhood sequence is padded if $S$ cannot be divided by $P$). Similarly, $\boldsymbol{X}_E$, $\boldsymbol{X}_T$ and $\boldsymbol{X}_M$ will be reshaped into $\mathbb{R}^{N_p \times (P \cdot d_E)}$, $\mathbb{R}^{N_p \times (P \cdot d_T)}$ and $\mathbb{R}^{N_p \times (P \cdot d_M)}$, respectively.

**Transformer Encoder.** Further, we apply a linear transformation to align the dimensions of various encodings. Specifically, given an encoding $\boldsymbol{X}_* \in \mathbb{R}^{N_p \times (P \cdot d_*)}$, we apply learnable weights $\boldsymbol{W}_* \in \mathbb{R}^{(P \cdot d_*) \times d}$ and bias $\boldsymbol{b}_* \in \mathbb{R}^d$ on it ($*$ can be $C, E, T, M$):

$$\boldsymbol{Z}_* = \boldsymbol{X}_* \boldsymbol{W}_* + \boldsymbol{b}_* \tag{10}$$

Then, we concatenate all these encodings $\boldsymbol{Z} = [\boldsymbol{Z}_C || \boldsymbol{Z}_E || \boldsymbol{Z}_T || \boldsymbol{Z}_M] \in \mathbb{R}^{N_p \times 4d}$. We set the input of HopeDGN as $\boldsymbol{H}^{(0)} = \boldsymbol{Z}$. The $l$-th layer ($1 \leq l \leq L$) of HopeDGN is defined as:

$$\begin{aligned}
\tilde{\boldsymbol{H}}^{(l)} &= \text{MHSA}(\text{LN}(\boldsymbol{H}^{(l-1)})) + \boldsymbol{H}^{(l-1)} \\
\boldsymbol{H}^{(l)} &= \text{FFN}(\text{LN}(\tilde{\boldsymbol{H}}^{(l)})) + \tilde{\boldsymbol{H}}^{(l)}
\end{aligned} \tag{11}$$

where MHSA, LN and FFN are the abbreviations of Multi-head Self-Attention, Layer Normalization and Feed-Forward Networks, respectively (Vaswani et al., 2017). The input and output dimensions of HopeDGN layer are set as same. After the final layer of Transformer encoder, the mean pooling with respect to the neighborhood is applied to obtain the embeddings of node pair $\boldsymbol{s} = (u, v)$ at $t$:

$$\boldsymbol{h}_t(\boldsymbol{s}) = \text{MEAN}(\boldsymbol{H}^{(L)}) \boldsymbol{W}_{out} + \boldsymbol{b}_{out} \in \mathbb{R}^{d_{out}} \tag{12}$$

where $\boldsymbol{W}_{out} \in \mathbb{R}^{4d \times d_{out}}$ and $\boldsymbol{b}_{out} \in \mathbb{R}^{d_{out}}$ are learnable weights.

**Computation complexity.** Given a batch of $B$ interactions, the expected cost of sampling the historical neighbors is $O(B \log(n_g))$, where $n_g$ is the average number of historical neighbors of temporal nodes in the dataset. Computing MITE costs $O(BS)$ since we traverse all the one-hop neighbors of the interactions in this batch. Forwarding propagation using the Transformer encoder costs $O(dS^2/P^2)$ since the input length has been reduced to $\lceil S/P \rceil$. Therefore, the overall complexity cost is $O(B(\log(n_g) + S) + dS^2/P^2)$. This complexity is same as the DyGFormer (Yu et al., 2023). We will compare the efficiency of HopeDGN and other baselines in Appendix D.4.

## 5 EXPERIMENTS

In this section, we conduct extensive experiments to evaluate the performance of the proposed Hope-DGN and MITE. Additional experiments are presented in Appendix D.

### 5.1 EXPERIMENTAL SETTINGS

**Datasets and Baselines.** Seven publicly available datasets are adopted for evaluation, namely, Reddit, Wikipedia, UCI, Enron, LastFM, MOOC and CanParl. These datasets are collected by Poursafaei et al. (2022). In addition, nine DyGNN baseline methods are leveraged for performance comparison, including JODIE (Kumar et al., 2019), DyRep (Trivedi et al., 2019), TGAT (Xu et al., 2020), TGN (Rossi et al., 2020), CAWN (Wang et al., 2021b), TCL (Wang et al., 2021a), PINT (Souza et al., 2022), GraphMixer (Cong et al., 2023) and DyGFormer (Yu et al., 2023).A detailed introduction to the datasets is presented in Appendix C.1.

**Evaluation Tasks and Metrics.** Our evaluation protocols closely follow Xu et al. (2020); Rossi et al. (2020). Specifically, we adopt future link prediction and temporal node classification tasks for evaluation. For the link prediction task, we randomly sample negative node pairs and train using the Binary Cross Entropy (BCE) loss function. The future link prediction task is divided into *transductive* and *inductive* settings. For both settings, we split the total time range $[0, T]$ into three time intervals $[0, 0.7T), [0.7T, 0.85T)$ and $[0.85T, T]$, and the interactions within each time interval formulate the training, validation and test sets, respectively. The model processes the interactions chronologically and predicts their existence based on interaction history. In the inductive setting, we randomly select 10% of nodes from the test set as *masking nodes*. The interactions associated with these masking nodes are removed during training, and the model is required to predict the interactions involving masking nodes only during the validation and testing phases. Average Precision (AP) and Area Under the Receiver Operating Characteristic curve (AUC) are used as evaluation metrics. The experimental settings of the dynamic node classification are presented in the Appendix D.2.

Table 1: The AP results of link prediction experiments. The values are multiplied by 100. The values of the best and second best performance are highlighted in **bold** and underlined, respectively.

| | Model | Reddit | Wikipedia | UCI | LastFM | Enron | MOOC | CanParl |
|---|---|---|---|---|---|---|---|---|
| Transductive | JODIE | 98.24±0.05 | 95.58±0.07 | 88.70±0.12 | 72.94±1.47 | 80.31±4.02 | 79.31±1.50 | 69.30±0.36 |
| | DyRep | 98.07±0.13 | 94.08±0.12 | 60.31±3.35 | 71.54±0.19 | 78.82±0.92 | 80.15±0.93 | 70.17±2.85 |
| | TGAT | 98.18±0.03 | 96.74±0.16 | 79.51±0.33 | 72.99±0.29 | 68.17±1.15 | 84.04±0.43 | 70.23±3.08 |
| | TGN | 98.62±0.02 | 98.15±0.07 | 90.25±0.26 | 77.15±2.01 | 85.73±1.60 | 87.76±0.21 | 68.82±1.01 |
| | CAWN | 99.11±0.00 | 98.77±0.02 | 94.92±0.01 | 89.15±0.00 | 88.92±0.19 | 79.78±0.16 | 71.13±1.50 |
| | GraphMixer | 96.89±0.00 | 96.67±0.04 | 92.97±0.68 | 75.63±0.15 | 82.24±0.01 | 81.96±0.11 | 77.53±0.11 |
| | TCL | 96.97±0.01 | 96.21±0.22 | 84.72±0.66 | 75.52±2.77 | 76.99±0.24 | 81.72±0.01 | 68.87±1.04 |
| | PINT | 99.03±0.01 | 98.78±0.01 | 96.01±0.10 | 88.06±0.70 | 88.71±1.30 | 71.54±2.62 | 68.39±0.10 |
| | DyGFormer | 99.22±0.01 | 98.96±0.00 | 95.43±0.14 | 91.90±0.04 | 92.20±0.12 | 85.63±0.34 | 97.35±0.33 |
| | HopeDGN | **99.31±0.01** | **99.17±0.03** | **97.18±0.06** | **93.16±0.03** | **92.67±0.08** | **90.19±0.29** | **98.33±0.60** |
| | Relative imprv.(%) | 0.09 | 0.21 | 1.22 | 1.37 | 0.51 | 2.77 | 1.01 |
| | Model | Reddit | Wikipedia | UCI | LastFM | Enron | MOOC | CanParl |
| Inductive | JODIE | 96.45±0.09 | 93.61±0.02 | 76.96±1.79 | 82.42±0.54 | 79.84±2.11 | 80.24±2.03 | 53.74±2.03 |
| | DyRep | 95.91±0.29 | 91.82±0.07 | 54.80±0.45 | 83.31±0.02 | 69.92±1.72 | 79.47±2.06 | 52.77±0.76 |
| | TGAT | 96.56±0.13 | 96.07±0.08 | 79.30±0.01 | 78.27±0.25 | 62.70±0.33 | 83.56±0.66 | 53.99±0.63 |
| | TGN | 97.35±0.09 | 97.51±0.03 | 84.26±1.31 | 85.02±0.96 | 78.71±3.99 | 87.37±0.26 | 54.43±2.92 |
| | CAWN | 98.65±0.02 | 98.20±0.02 | 92.29±0.12 | 91.50±0.00 | 85.26±0.15 | 80.98±0.16 | 55.64±0.23 |
| | GraphMixer | 94.93±0.01 | 96.11±0.07 | 90.82±0.58 | 82.22±0.32 | 75.42±0.08 | 80.34±0.21 | 56.19±0.52 |
| | TCL | 94.03±0.35 | 96.11±0.14 | 82.53±0.77 | 80.43±2.40 | 73.43±0.25 | 79.98±0.06 | 54.25±0.51 |
| | PINT | 98.25±0.04 | 98.38±0.04 | 93.97±0.10 | 91.76±0.70 | 81.05±2.40 | 73.10±2.92 | 49.57±1.57 |
| | DyGFormer | 98.83±0.02 | 98.53±0.04 | 93.66±0.13 | 93.29±0.02 | 89.57±0.16 | 85.04±0.33 | 86.79±2.31 |
| | HopeDGN | **98.97±0.01** | **98.87±0.00** | **95.56±0.05** | **94.45±0.08** | **89.84±0.02** | **90.10±0.34** | **88.84±0.74** |
| | Relative imprv.(%) | 0.15 | 0.34 | 1.69 | 1.24 | 0.30 | 3.12 | 2.35 |

**Model Configurations.** We train the proposed HopeDGN and other baseline methods for 50 epochs using the early stopping strategies of patience of 10. The model achieving the best performance on validation set is selected for testing. For all models, the optimizer, learning rate and batch size are set as 0.0001, 200 and Adam (Kingma & Ba, 2014), respectively. We repeat the experiments three times with different random seeds and report the mean and standard deviation results. Other configurations of HopeDGN and baseline methods are presented in Appendix C.2.

## 5.2 RESULTS AND DISCUSSION

The AP results of the proposed HopeDGN and other baselines on the link prediction task are presented in Table 1. The AUC results are presented in Table 5. From Table 1, we have following observations. Firstly, under both transductive and inductive settings, the proposed HopeDGN achieves the best performance on all datasets among the eight baseline methods. Specifically, the proposed HopeDGN achieves an average AP improvement of 1.02% and 1.31% for transductive and inductive experiments over the second-best baselines, respectively. These results demonstrate the effectiveness of HopeDGN. Secondly, the MITE used in HopeDGN is a generalized form of NCOE used in DyGFormer. Compared to DyGFormer, the proposed HopeDGN shows an improvement of 4.56% in the transductive setting and 5.06% in the inductive setting on the MOOC dataset. This may be because the proposed HopeDGN leverages MITE, which encodes the complete bi-interaction history to the target node pairs, thus enriching more semantic information than NCOE used in DyGFormer. The experimental results of the dynamic node classification are presented in the Appendix D.2.

## 5.3 ABLATION STUDIES

In this section, we conduct ablation studies to validate the effectiveness of key components of Hope-DGN , including MITE and Time Encoding (TE). We respectively remove MITE (denoted as "w/o MITE") and TE (denoted as "w/o TE"), and compare their performance with original model. The evaluating datasets include UCI, CanParl, Enron and MOOC. The results are presented in Fig. 2. From Fig. 2, we observe that the MITE plays the most significant role in the performance of Hope-DGN, as removing this module causes significant performance drop. In addition, time encoding is vital for some datasets such as CanParl and MOOC.

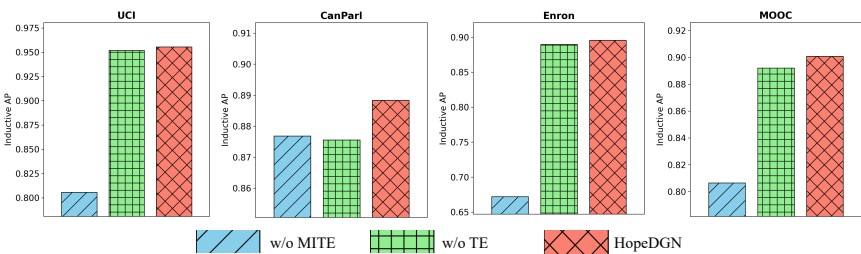

Figure 2: Ablation studies on the components of HopeDGN.

## 5.4 Incorporating MITE with other baselines

To validate the flexibility of the proposed MITE, we evaluate the performance of incorporating MITE with other baselines using link prediction tasks. The AP results are presented in Table 2. From Table 2, we observe that the performance of the baseline models significantly boosts after incorporating MITE. In particular, on the Enron dataset, the performance of TGAT improves by 33.21% and 39.23% for transductive and inductive settings, respectively, after incorporating MITE encoding. The reason may be that MITE can provide the indirect dependency information of node pairs as additional feature information, which is helpful for link prediction tasks. Note that the proposed HopeDGN still achieves the best performance among all the baselines incorporating MITE.

Table 2: The performance of incorporating MITE with baselines on link prediction tasks. The values are multiplied by 100. The values of the best performance are highlighted in **bold**.

|  | Transductive AP | | | Inductive AP | | |
|---|---|---|---|---|---|---|
|  | LastFM | Enron | MOOC | LastFM | Enron | MOOC |
| TGAT | 72.99±0.29 | 68.17±1.15 | 84.04±0.43 | 78.27±0.25 | 62.70±0.33 | 83.56±0.66 |
| TGAT w/ MITE | 89.32±0.14 | 90.81±0.15 | 88.31±0.29 | 91.75±0.19 | 87.30±0.43 | 87.72±0.33 |
| Relative Imprv. (%) | 22.37 | 33.21 | 5.08 | 17.22 | 39.23 | 4.97 |
| Graphmixer | 75.63±0.15 | 82.24±0.01 | 81.96±0.11 | 82.22±0.32 | 75.42±0.08 | 80.34±0.21 |
| Graphmixer w/ MITE | 86.59±0.03 | 91.30±0.06 | 87.30±0.13 | 89.98±0.00 | 88.71±0.13 | 86.22±0.31 |
| Relative Imprv. (%) | 14.49 | 11.01 | 6.51 | 9.43 | 17.62 | 7.31 |
| TCL | 75.52±2.77 | 76.99±0.24 | 81.72±0.01 | 80.43±2.40 | 73.43±0.25 | 79.98±0.06 |
| TCL w/ MITE | 89.60±0.01 | 90.53±0.14 | 88.36±0.01 | 92.23±0.00 | 88.17±0.08 | 87.44±0.09 |
| Relative Imprv. (%) | 18.64 | 17.58 | 8.12 | 14.67 | 20.07 | 9.32 |
| HopeDGN | **93.16±0.03** | **92.67±0.08** | **90.19±0.29** | **94.45±0.08** | **89.58±0.20** | **90.10±0.34** |

## 6 Conclusions

In this work, we propose a novel DyGNN framework that can achieve provable and quantifiable high-order expressive power. We propose the $k$-Dynamic WL (DWL) tests to quantify the expressive power of DyGNNs. We underscore that the expressive power of existing DyGNNs is bounded by the proposed 1-DWL test, which limits their capabilities to capture significant evolving patterns. To address this limitation, we propose HopeDGN, which learns node-pair level representation by aggregating interaction histories with neighboring node-pairs. We prove that HopeDGN can achieve expressive power equivalent to the 2-DWL test. We present a Transformer-based implementation for the local variant HopeDGN. Experiments on link prediction and node classification tasks demonstrate the effectiveness of HopeDGN.

There are some promising future directions for this work. Firstly, the expressive power of the proposed HopeDGN is bounded by 2-DWL test. It remains an open problem of designing DyGNNs with higher order expressiveness than 2-DWL test. Secondly, some recent works study the expressiveness of GNNs via alternative metrics beyond WL test such as graph bi-connectivity (Zhang et al., 2023a). Designing DyGNNs based on other metrics beyond DWL test may bring novel insights.

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

# A    AN INTRODUCTION OF WEISFEILER-LEHMAN TEST

In this section, we briefly introduce the Weisfeiler-Lehman (WL) test. The WL tests for graph isomorphism (Leman & Weisfeiler, 1968; Xu et al., 2019) are effective algorithms that have been proven capable of discriminating a broad class of non-isomorphic graphs.

**1-WL test.**    Given a $\mathcal{G} = \{\mathcal{V}, \mathcal{E}\}$ with a labeling function $l$, at iteration 0, the 1-WL test initializes the color of each node $c^{(0)} = l$. At iteration $j > 0$, the node color is refined as:

$$c^{(j)}(u) = \text{HASH}(c^{(j-1)}(u), \{\!\!\{c^{(j-1)}(v) | v \in \mathcal{N}(u)\}\!\!\}) \tag{13}$$

where HASH is a hashing function and $\{\!\!\{\cdot\}\!\!\}$ denotes the multiset. To test whether two graphs $\mathcal{G}$ and $\mathcal{G}'$ are isomorphic, we run 1-WL test on both graphs in parallel. If the multisets of node colors in two graphs are not equal at any iteration, the 1-WL test concludes that $\mathcal{G}$ and $\mathcal{G}'$ are non-isomorphic.

Due to the limited expressive power of 1-WL test, the $k$-dimensional Weisfeiler-Lehman tests ($k \geq 2$) are proposed to serve as more powerful algorithms for checking graph isomorphism. In the literature, there exists two variants of $k$-WL test, known as Folklore $k$-WL test ($k$-FWL) (Cai et al., 1992) and Oblivious $k$-WL test ($k$-OWL) (Grohe, 2021). We will introduce the details of them.

**$k$-FWL test.**    Given a graph $\mathcal{G} = \{\mathcal{V}, \mathcal{E}\}$, and let $\boldsymbol{s} = (v_1, ..., v_k) \in [\mathcal{V}]^k$ be a $k$-node tuple. Let $c^{(j)} : [\mathcal{V}]^k \to \mathbb{N}$ be a $k$-node tuple coloring function at iteration $j$. At iteration 0, two tuples $\boldsymbol{s}$ and $\boldsymbol{s}'$ get the same color if there exists a isomorphism between $\boldsymbol{s}$ and $\boldsymbol{s}'$. Then at iteration $j$ ($j \geq 1$), the color of $\boldsymbol{s}$ is updated as follows:

$$c^{(j)}(\boldsymbol{s}) = \text{HASH}\big(c^{(j-1)}(\boldsymbol{s}), \{\!\!\{\boldsymbol{\phi}^{(j-1)}(\boldsymbol{s}, w) | w \in \mathcal{V}\}\!\!\}\big)$$
$$\boldsymbol{\phi}^{(j-1)}(\boldsymbol{s}, w) = \Big(c^{(j-1)}\big(\boldsymbol{r}_1(\boldsymbol{s}, w)\big), ..., c^{(j-1)}\big(\boldsymbol{r}_k(\boldsymbol{s}, w)\big)\Big) \tag{14}$$

where $\boldsymbol{r}_i(\boldsymbol{s}, w) = (v_1, ..., v_{i-1}, w, v_{i+1}, ..., v_k)$. Here the neighboring node tuples of $s$ is obtained by replacing each element in $\boldsymbol{s}$ with other nodes. We run the algorithm on two graphs in parallel. If two color multisets are not equal at any iteration, the $k$-FWL test will output that these two graphs are non-isomorphic. The algorithm terminates if $c^{(j)}(\boldsymbol{s}) = c^{(j)}(\boldsymbol{s}') \iff c^{(j+1)}(\boldsymbol{s}) = c^{(j+1)}(\boldsymbol{s}')$ holds for all $\boldsymbol{s}, \boldsymbol{s}' \in [\mathcal{V}]^k$.

**$k$-OWL test.**    At iteration $j$ ($j \geq 1$), $k$-OWL test has a slightly different update rule for the colors of $\boldsymbol{s} \in [\mathcal{V}]^k$:

$$c^{(j)}(\boldsymbol{s}) = \text{HASH}\big(c^{(j-1)}(\boldsymbol{s}), M^{(j-1)}(\boldsymbol{s})\big)$$
$$M^{(j-1)}(\boldsymbol{s}) = \Big(\{\!\!\{c^{(j-1)}\big(\boldsymbol{r}_1(\boldsymbol{s}, w)\big) | w \in \mathcal{V}\}\!\!\}, ..., \{\!\!\{c^{(j-1)}\big(\boldsymbol{r}_k(\boldsymbol{s}, w)\big) | w \in \mathcal{V}\}\!\!\}\Big) \tag{15}$$

Note that 1-OWL test and 2-OWL test have the same expressive power, and $(k + 1)$-OWL test has the same expressive power as the $k$-FWL test for $k \geq 2$ (Grohe, 2021). The reason why $k$-FWL test inherits more expressive power than $k$-OWL test is that $k$-FWL firstly groups the color of $k$-node tuple based on replacing nodes then makes aggregation, while $k$-OWL aggregates the colors for the single replacing node.

# B    PROOF OF PROPOSITIONS

## B.1    PROOF OF PROPOSITION 1

We restate Proposition 1 as follows.

**Proposition 6.** *Let $\mathcal{DG} = \{\mathcal{V}, \mathcal{E}\}$ and $\mathcal{DG}' = \{\mathcal{V}', \mathcal{E}'\}$ be two dynamic graphs. Suppose the initial labeling function of $k$-DWL test be constant. Then, for all $k \geq 1$, if $k$-DWL test decides $\mathcal{DG}$ and $\mathcal{DG}'$ are non-isomorphic, then $(k + 1)$-DWL test also decides $\mathcal{DG}$ and $\mathcal{DG}'$ are non-isomorphic.*

*Proof.* Let $\boldsymbol{s}_k \in [\mathcal{V}]^k$ and $\boldsymbol{s}'_k \in [\mathcal{V}']^k$ be the $k$-node tuples on $\mathcal{DG}$ and $\mathcal{DG}'$, respectively. We use $c_k^{(i)}(\boldsymbol{s}_k)$ to denote the color of $\boldsymbol{s}_k$ at the $i$-th iteration of $k$-DWL test.

Suppose after $j$ iterations, $k$-DWL test determines $\mathcal{DG}$ and $\mathcal{DG}'$ are non-isomorphic, but $(k+1)$-DWL test determines $\mathcal{DG}$ and $\mathcal{DG}'$ are isomorphic. It follows that from iteration $i = 0, 1, ..., j$, $\{\!\{c_{k+1}^{(i)}(\boldsymbol{s}_{k+1})|\boldsymbol{s}_{k+1} \in [\mathcal{V}]^{k+1}\}\!\} = \{\!\{c_{k+1}^{(i)}(\boldsymbol{s}'_{k+1})|\boldsymbol{s}'_{k+1} \in [\mathcal{V}']^{k+1}\}\!\}$. Let $\boldsymbol{s}_{k+1} = (v_1, ..., v_k, v_{k+1})$ and $\boldsymbol{s}'_{k+1} = (v'_1, ..., v'_k, v'_{k+1})$. We will show that for $i = 0, ..., j$, $c_{k+1}^{(i)}(\boldsymbol{s}_{k+1}) = c_{k+1}^{(i)}(\boldsymbol{s}'_{k+1}) \implies c_k^{(i)}((v_1, ..., v_k)) = c_k^{(i)}((v'_1, ..., v'_k))$. We prove this by induction on the iteration $i$.

[Base Case]: For $i = 0$, $c_{k+1}^{(0)}(\boldsymbol{s}_{k+1}) = c_{k+1}^{(0)}(\boldsymbol{s}'_{k+1}) \implies c_k^{(0)}((v_1, ..., v_k)) = c_k^{(0)}((v'_1, ..., v'_k))$ immediately holds because the initial labeling function of $k$ and $(k+1)$-DWL tests are constant.

[Inductive Step]: Suppose $c_{k+1}^{(i)}(\boldsymbol{s}_{k+1}) = c_{k+1}^{(i)}(\boldsymbol{s}'_{k+1}) \implies c_k^{(i)}((v_1, ..., v_k)) = c_k^{(i)}((v'_1, ..., v'_k))$ holds for iteration $i$. Then, for iteration $i+1$, we discuss by cases:

- $k = 1$. Based on Eq. (5), $c_{k+1}^{(i+1)}(\boldsymbol{s}_{k+1}) = c_{k+1}^{(i+1)}(\boldsymbol{s}'_{k+1})$ implies: 1) $c_{k+1}^{(i)}(\boldsymbol{s}_{k+1}) = c_{k+1}^{(i)}(\boldsymbol{s}'_{k+1})$. Based on the induction assumption, this implies:

$$c_k^{(i)}(v_1) = c_k^{(i)}(v'_1) \tag{16}$$

  2) $\{\!\{\phi_t^{(i-1)}(\boldsymbol{s}_{k+1}, w)|w \in \mathcal{V}\}\!\} = \{\!\{\phi_t^{(i-1)}(\boldsymbol{s}'_{k+1}, w')|w' \in \mathcal{V}'\}\!\}$, which implies that:

$$\{\!\{(c_{k+1}^{(i-1)}(w, v_1), \mathbf{A}_{w,v_1,:})|w \in \mathcal{V}\}\!\} = \{\!\{(c_{k+1}^{(i-1)}(w', v'_1), \mathbf{A}_{w',v'_1,:})|w' \in \mathcal{V}'\}\!\} \tag{17}$$

  Based on the induction assumption, this implies:

$$\{\!\{(c_k^{(i-1)}(w), \mathbf{A}_{w,v_1,:})|w \in \mathcal{V}\}\!\} = \{\!\{(c_k^{(i-1)}(w', v'_1), \mathbf{A}_{w',v'_1,:})|w' \in \mathcal{V}'\}\!\} \tag{18}$$

  This implies:

$$\{\!\{(c_k^{(i-1)}(w), \mathbf{A}_{w,v_1,:})|w \in \mathcal{N}(v_1)\}\!\} = \{\!\{(c_k^{(i-1)}(w', v'_1), \mathbf{A}_{w',v'_1,:})|w' \in \mathcal{N}(v'_1)\}\!\} \tag{19}$$

  where $\mathcal{N}(v_1) = \{\!\{w|w \in \mathcal{V}, \mathbf{A}_{w,v_1,:} \neq [\infty]\}\!\}$ indicates the nodes that have interactions with $v_1$. Then based on Eq. (4), combining Eq. (16) and Eq. (19) yields $c_k^{(i+1)}(v_1) = c_k^{(i+1)}(v'_1)$

- $k > 1$. Based on Eq. (5), $c_{k+1}^{(i+1)}(\boldsymbol{s}_{k+1}) = c_{k+1}^{(i+1)}(\boldsymbol{s}'_{k+1})$ implies: 1) $c_{k+1}^{(i)}(\boldsymbol{s}_{k+1}) = c_{k+1}^{(i)}(\boldsymbol{s}'_{k+1})$. Based on the induction assumption, this implies:

$$c_k^{(i)}((v_1, ..., v_k)) = c_k^{(i)}((v'_1, ..., v'_k)) \tag{20}$$

  2) $\{\!\{\phi_t^{(i-1)}(\boldsymbol{s}_{k+1}, w)|w \in \mathcal{V}\}\!\} = \{\!\{\phi_t^{(i-1)}(\boldsymbol{s}'_{k+1}, w')|w' \in \mathcal{V}'\}\!\}$, which implies that:

$$\{\!\{(c_{k+1}^{(i-1)}((w, v_2, ..., v_{k+1})), ..., c_{k+1}^{(i-1)}((v_1, ..., v_k, w)), \mathbf{A}_{w,v_1,:}, ..., \mathbf{A}_{w,v_{k+1},:})|w \in \mathcal{V}\}\!\}$$
$$= \{\!\{(c_{k+1}^{(i-1)}((w', v'_2, ..., v'_{k+1})), ..., c_{k+1}^{(i-1)}((v'_1, ..., v'_k, w')), \mathbf{A}_{w',v'_1,:}, ..., \mathbf{A}_{w',v'_{k+1},:})|$$
$$w' \in \mathcal{V}'\}\!\} \tag{21}$$

  Based on the induction assumption, this implies:

$$\{\!\{(c_k^{(i-1)}(w, v_2..., v_k), c_k^{(i-1)}(v_1, w..., v_k), ..., c_k^{(i-1)}(v_1, ..., v_{k-1}, w),$$
$$\mathbf{A}_{w,v_1,:}, ..., \mathbf{A}_{w,v_k,:})|w \in \mathcal{V}\}\!\}$$
$$= \{\!\{(c_k^{(i-1)}(w', v'_2..., v'_k), c_k^{(i-1)}(v'_1, w'..., v'_k), ..., c_k^{(i-1)}(v'_1, ..., v'_{k-1}, w'),$$
$$\mathbf{A}_{w',v'_1,:}, ..., \mathbf{A}_{w',v'_k,:})|w' \in \mathcal{V}'\}\!\} \tag{22}$$

  Then based on Eq.(5), combining Eq.(20) and Eq.(22), yields $c_k^{(i)}((v_1, ..., v_k)) = c_k^{(i)}((v'_1, ..., v'_k))$

Combining the base case and inductive step, it holds that for $i = 0, ..., j$, $c_{k+1}^{(i)}(\boldsymbol{s}_{k+1}) = c_{k+1}^{(i)}(\boldsymbol{s}'_{k+1}) \implies c_k^{(i)}((v_1, ..., v_k)) = c_k^{(i)}((v'_1, ..., v'_k))$. We use $g$ to denote the mapping $(v_1, ..., v_k) = g(\boldsymbol{s}_{k+1})$.. Then, from iteration $i = 0, ..., j$, it holds

$$
\begin{aligned}
&\{\!\{c_{k+1}^{(i)}(\boldsymbol{s}_{k+1})|\boldsymbol{s}_{k+1} \in [\mathcal{V}]^{k+1}\}\!\} = \{\!\{c_{k+1}^{(i)}(\boldsymbol{s}'_{k+1})|\boldsymbol{s}'_{k+1} \in [\mathcal{V}']^{k+1}\}\!\} \\
\implies &\{\!\{c_k^{(i)}(g(\boldsymbol{s}_{k+1}))|\boldsymbol{s}_{k+1} \in [\mathcal{V}]^{k+1}\}\!\} = \{\!\{c_{k+1}^{(i)}(g(\boldsymbol{s}'_{k+1}))|\boldsymbol{s}'_{k+1} \in [\mathcal{V}']^{k+1}\}\!\} \\
\implies &\{\!\{c_{k+1}^{(i)}(g(\boldsymbol{s}_k))|\boldsymbol{s}_k \in [\mathcal{V}]^k\}\!\} = \{\!\{c_k^{(i)}(g(\boldsymbol{s}'_k))|\boldsymbol{s}'_k \in [\mathcal{V}']^k\}\!\}
\end{aligned}
\tag{23}
$$

This indicates that $k$-DWL test concludes that $\mathcal{DG}$ and $\mathcal{DG}'$ are isomorphic. This causes the contradiction. Thus, the proposition is proved.

$\square$

## B.2 PROOF OF PROPOSITION 2

We restate Proposition 2 as follows.

**Proposition 7.** *Let $\mathcal{DG} = \{\mathcal{V}, \mathcal{E}\}$ and $\mathcal{DG}' = \{\mathcal{V}', \mathcal{E}'\}$ be two dynamic graphs, and $\boldsymbol{X}$ and $\boldsymbol{X}'$ be their corresponding node features. Given a node labeling function $l : \mathcal{V} \to \mathbb{N}$ satisfying $l(u) = l(v)$ if and only if $\boldsymbol{X}_u = \boldsymbol{X}'_v$ for any $u \in \mathcal{V}$ and $v \in \mathcal{V}'$. Let $c_t^{(j)}$ denotes the color at time $t$ obtained by 1-DWL test initialized with label function $l$ in the $j$-th iteration, and $\boldsymbol{h}_t^{(j)}$ be the temporal node embeddings outputted by the DyGNN. Then for all $j \geq 0$, $c_t^{(j)}(u) = c_t^{(j)}(v) \implies \boldsymbol{h}_t^{(j)}(u) = \boldsymbol{h}_t^{(j)}(v)$.*

*Proof.* We prove this proposition by induction on the iteration $j$.

[Base Case]: For $j = 0$, we have:

$$
c_t^{(0)}(u) = c_t^{(0)}(v) \overset{(a)}{\implies} l(u) = l(v) \overset{(b)}{\implies} \boldsymbol{X}_u = \boldsymbol{X}'_v \implies \boldsymbol{h}_t^{(0)}(u) = \boldsymbol{h}_t^{(0)}(u)
\tag{24}
$$

where $(a)$ is because 1-DWL test is initialized with $l$. $(b)$ is due to the consistency assumption of $l$.

[Inductive Step]: Suppose $c_t^{(j)}(u) = c_t^{(j)}(v) \implies \boldsymbol{h}_t^{(j)}(u) = \boldsymbol{h}_t^{(j)}(v)$ holds for iteration $j$. Then, based on Eq. (4), at iteration $j + 1$, $c_t^{(j+1)}(u) = c_t^{(j+1)}(v)$ implies: 1) $c_t^{(j)}(u) = c_t^{(j)}(v)$. 2) $\{\!\{(c_t^{(j)}(w), \mathbf{A}_{w,u,:}^{<t})|(w, t') \in \mathcal{N}(u, t)\}\!\} = \{\!\{(c_t^{(j)}(r), \mathbf{A}_{r,v,:}^{<t})|(r, t') \in \mathcal{N}(v, t)\}\!\}$. Then we have:

$$
c_t^{(j)}(u) = c_t^{(j)}(v) \implies \boldsymbol{h}_t^{(j)}(u) = \boldsymbol{h}_t^{(j)}(v)
\tag{25}
$$

due to the inductive assumption. In addition,

$$
\begin{aligned}
&\{\!\{(c_t^{(j)}(w), \mathbf{A}_{w,u,:}^{<t})|(w, t') \in \mathcal{N}(u, t)\}\!\} = \{\!\{(c_t^{(j)}(r), \mathbf{A}_{r,v,:}^{<t})|(r, t') \in \mathcal{N}(v, t)\}\!\} \\
\implies &\{\!\{(\boldsymbol{h}_t^{(j)}(w), \mathbf{A}_{w,u,:}^{<t})|(w, t') \in \mathcal{N}(u, t)\}\!\} = \{\!\{(\boldsymbol{h}_t^{(j)}(r), \mathbf{A}_{r,v,:}^{<t})|(r, t') \in \mathcal{N}(v, t)\}\!\} \\
\implies &\{\!\{(\boldsymbol{h}_t^{(j)}(w), t - t')|(w, t') \in \mathcal{N}(u, t)\}\!\} = \{\!\{(\boldsymbol{h}_t^{(j)}(r), t - t')|(r, t') \in \mathcal{N}(v, t)\}\!\}
\end{aligned}
\tag{26}
$$

where the last equation holds because the entire interaction sequence being the same implies that each interaction time is the same. Combining Eq. (25) and Eq. (26), and based on Eq. (1), we have $\boldsymbol{h}_t^{(j+1)}(u) = \boldsymbol{h}_t^{(j+1)}(v)$, which concludes the proof. $\square$

## B.3 PROOF OF PROPOSITION 3

We restate Proposition 3 as follows.

**Proposition 8.** *There exists two dynamic graphs $\mathcal{DG} = \{\mathcal{V}, \mathcal{E}\}$ and $\mathcal{DG}' = \{\mathcal{V}', \mathcal{E}'\}$ which have non-isomorphic node pairs $\boldsymbol{s} \in [\mathcal{V}]^2$ and $\boldsymbol{s}' \in [\mathcal{V}']^2$ until some time $t$ that DyGNN with MITE can distinguish while vanilla DyGNN cannot.*

*Proof.* We present a case of non-isomorphic node pairs that DyGNN with MITE can distinguish while vanilla DyGNN cannot in Fig. 3. In Fig. 3, suppose the current time is $t_5$. It can be seen that node pair $(a, c)$ in $\mathcal{DG}$ (a) is not isomorphic to the node pair $(a, g)$ in $\mathcal{DG}$ (b) until $t_5$, because the

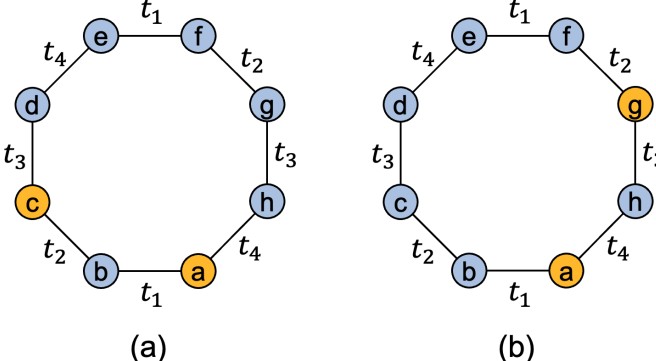

Figure 3: An example of Proposition 3. Suppose the raw node feature are same for all nodes, and the current time is $t_5$. The model is required to distinguish two node pairs $(a, c)$ in (a) and $(a, g)$ in (b) at time $t_5$.

common neighbor $b$ interacts with $(a, c)$ at time $t_1$ and $t_2$ in $\mathcal{DG}$ (a), while the common neighbor $h$ interacts with $(a, g)$ at time $t_3$ and $t_4$ in $\mathcal{DG}$ (b).

For vanilla DyGNN, we denote the temporal node embeddings of $\mathcal{DG}$ (a) and (b) are $\boldsymbol{h}$ and $\boldsymbol{q}$, respectively. Then we have $\boldsymbol{h}_{t_5}^{(l)}(a) = \boldsymbol{q}_{t_5}^{(l)}(a)$ and $\boldsymbol{h}_{t_5}^{(l)}(c) = \boldsymbol{q}_{t_5}^{(l)}(g)$ for any $l \geq 0$, because the corresponding nodes are isomorphic. The node pair embedding of $(a, c)$ in $\mathcal{DG}$ (a) at $t_5$ is $[\boldsymbol{h}_{t_5}^{(l)}(a) || \boldsymbol{h}_{t_5}^{(l)}(c)]$, and the node pair embedding of $(a, g)$ in $\mathcal{DG}$ (b) at $t_5$ is $[\boldsymbol{q}_{t_5}^{(l)}(a) || \boldsymbol{q}_{t_5}^{(l)}(g)]$. Therefore, vanilla DyGNN cannot distinguish this two node pairs.

For DyGNN with MITE. we note that MITE of $b$ with respect to $(a, c)$ is $[t_5 - t_1 || t_5 - t_2]$ in $\mathcal{DG}$ (a), while MITE of $h$ with respect to $(a, g)$ is $[t_5 - t_4 || t_5 - t_3]$ in $\mathcal{DG}$ (b). Therefore, when MITE is concatenated with the raw node feature, aggregating neighbor information of node pair $(a, c)$ in $\mathcal{DG}$ (a) and node pair $(a, g)$ in $\mathcal{DG}$ (b) will yield $\boldsymbol{h}_{t_5}^{(l)}(a) \neq \boldsymbol{q}_{t_5}^{(l)}(a)$ and $\boldsymbol{h}_{t_5}^{(l)}(c) \neq \boldsymbol{q}_{t_5}^{(l)}(g)$. Therefore, DyGNN with MITE can distinguish this two node pairs. $\qquad\square$

### B.4 PROOF OF PROPOSITION 4

We restate Proposition 4 as follows.

**Proposition 9.** *Let $\mathcal{DG} = \{\mathcal{V}, \mathcal{E}\}$ and $\mathcal{DG}' = \{\mathcal{V}', \mathcal{E}'\}$ be two dynamic graphs, and $\boldsymbol{X}$ and $\boldsymbol{X}'$ be their corresponding node features. Given a node labeling function $l : [\mathcal{V}]^2 \to \mathbb{N}$ satisfying $l((u, v)) = l((u', v'))$ if and only if $[\boldsymbol{X}_u || \boldsymbol{X}_v] = [\boldsymbol{X}'_{u'} || \boldsymbol{X}'_{v'}]$ for all $(u, v) \in [\mathcal{V}]^2$ and $(u', v') \in [\mathcal{V}']^2$. Let $c_t^{(j)}$ denotes the color at time $t$ obtained by 2-DWL test , initialized with label function $l$ in the $j$-th iteration, and $\boldsymbol{h}_t^{(j)}$ be the temporal node embeddings output by the Global HopeDGN. Then for all $j \geq 0$, $c_t^{(j)}\big((u, v)\big) = c_t^{(j)}\big((u', v')\big) \implies \boldsymbol{h}_t^{(j)}\big((u, v)\big) = \boldsymbol{h}_t^{(j)}\big((u', v')\big)$.*

*Proof.* We prove this proposition by induction on the iteration $j$.

[Base Case]: For $j = 0$, we have:

$$c_t^{(0)}((u, v)) = c_t^{(0)}((u', v')) \overset{(a)}{\implies} l((u, v)) = l((u', v')) \overset{(b)}{\implies} [\boldsymbol{X}_u || \boldsymbol{X}_v] = [\boldsymbol{X}'_{u'} || \boldsymbol{X}'_{v'}]$$
$$\implies \boldsymbol{h}_t^{(0)}((u, v)) = \boldsymbol{h}_t^{(0)}((u', v')) \tag{27}$$

where $(a)$ is because we assign $l$ as the initial coloring of 2-DWL and $(b)$ is due to the consistency assumption of $l$.

[Inductive Step]: Suppose $c_t^{(j)}((u, v)) = c_t^{(j)}((u', v')) \implies \boldsymbol{h}_t^{(j)}((u, v)) = \boldsymbol{h}_t^{(j)}((u', v'))$ holds for iteration $j$. Then, because of Eq. (5), $c_t^{(j+1)}((u, v)) = c_t^{(j+1)}((u', v'))$ implies: 1) $c_t^{(j)}((u, v)) =$

$c_t^{(j)}((u', v'))$, 2) $\{\{\phi_t^{(j)}((u, v), w) | w \in \mathcal{V}\}\} = \{\{\phi_t^{\boldsymbol{(j)}}((u', v'), w') | w' \in \mathcal{V}'\}\}$. Based on the inductive assumption, we have

$$c_t^{(j)}((u, v)) = c_t^{(j)}((u', v')) \implies \boldsymbol{h}_t^{(j)}((u, v)) = \boldsymbol{h}_t^{(j)}((u', v')) \tag{28}$$

Also, there exists a mapping $f : \mathcal{V} \to \mathcal{V}'$ where $\phi_t^{(j)}((u, v), w) = \phi_t^{(j)}((u', v'), f(w))$. This means that $c_t^{(j)}((u, w)) = c_t^{(j)}((u', f(w))), (c_t^{(j)}((v, w)) = c_t^{(j)}((v', f(w))$ and $\mathbf{A}_{w,u,:}^{<t} = \mathbf{A}_{f(w),u',:}^{<t}, \mathbf{A}_{w,v,:}^{<t} = \mathbf{A}_{f(w),v',:}^{<t}$ holds for any $w \in \mathcal{V}$. This indicates the followings:

- $\forall w \in \mathcal{V}, [\boldsymbol{h}_t^{(j)}((u, w)) \, || \, \boldsymbol{h}_t^{(j)}((v, w))] = [\boldsymbol{h}_t^{(j)}((u', f(w))) \, || \, \boldsymbol{h}_t^{(j)}((v', f(w)))]$ by the inductive assumption on iteration $j$.

- $\forall w \in \mathcal{V}, [\mathbf{B}_{u,w,:}^t \, || \, \mathbf{B}_{v,w,:}^t] = [\mathbf{B}_{u',f(w),:}^t \, || \, \mathbf{B}_{v',f(w),:}^t]$. This is because all the interaction times between node pairs $(u, w)$ and $(u', f(w))$ before time $t$ are the same, they thus generate the same TITs, and the same holds for node pairs $(v, w)$ and $(v', f(w))$.

Combining the above facts and Eq. (28), and based on Eq. (8), we have $\boldsymbol{h}_t^{(j+1)}((u, v)) = \boldsymbol{h}_t^{(j+1)}((u', v'))$, which concludes the proof. $\square$

### B.5 PROOF OF PROPOSITION 5

We restate Proposition 5 as follows.

**Proposition 10.** *Let* $\mathcal{M} : [\mathcal{V}]^2 \to \mathbb{R}^d$ *be a Global HopeDGN. Suppose the 2-DWL test is initialized with a node labeling function* $l : [\mathcal{V}]^2 \to \mathbb{N}$ *satisfying* $l((u, v)) = l((u', v'))$ *if and only if* $[\boldsymbol{X}_u || \boldsymbol{X}_v] = [\boldsymbol{X}_{u'}' || \boldsymbol{X}_{v'}']$ *for all* $(u, v) \in [\mathcal{V}]^2$ *and* $(u', v') \in [\mathcal{V}']^2$. *If the* AGG, UPDATE, $f_1$ *and* $f_2$ *of* $\mathcal{M}$ *are injective, then at any time* $t$, *if 2-DWL test assigns different colors to two node pairs,* $\mathcal{M}$ *will also output different temporal embeddings of these two node pairs.*

*Proof.* We will show that for all $j \geq 0$, there exists an injective function $\varphi$ where for all $(u, v) \in [\mathcal{V}]^2, \boldsymbol{h}_t^{(j)}((u, v)) = \varphi(c_t^{(j)}((u, v)))$ holds. We prove this by induction on $j$.

[Base Case]: When $j = 0$, considering the consistency assumption of node labeling function $l$ and combined node features, we have:

$$\begin{aligned} c_t^{(0)}((u, v)) \neq c_t^{(0)}((u', v')) &\implies l((u, v)) \neq l((u', v')) \implies [\boldsymbol{X}_u || \boldsymbol{X}_v] \neq [\boldsymbol{X}_{u'}' || \boldsymbol{X}_{v'}'] \\ &\implies \boldsymbol{h}_t^{(0)}((u, v)) \neq \boldsymbol{h}_t^{(0)}((u', v')) \end{aligned} \tag{29}$$

thus by the definition of injectiveness, there must exists an injective function $\varphi$ such that $\boldsymbol{h}_t^{(0)}((u, v)) = \varphi(c_t^{(0)}((u, v)))$.

[Inductive Case]: Suppose for iteration $j$, and suppose $\varphi$ is the injective function satisfying $\boldsymbol{h}_t^{(j)}((u, v)) = \varphi(c_t^{(j)}((u, v)))$. Then for iteration $j + 1$, we have

$$\begin{aligned} \boldsymbol{h}_t^{(j+1)}((u, v)) &= \text{UPDATE}(\boldsymbol{h}_t^{(j)}((u, v)), \text{AGG}(\{\{[f_1([\boldsymbol{h}_t^{(j)}((u, w)) \, || \, \boldsymbol{h}_t^{(j)}((v, w))]) \, || \\ &\qquad\qquad f_2([\mathbf{B}_{u,w,:}^t \, || \, \mathbf{B}_{v,w,:}^t])] \, | \, w \in \mathcal{V}\}\}) \\ &= \text{UPDATE}(\varphi(c_t^{(j)}((u, v))), \text{AGG}(\{\{[f_1([\varphi(c_t^{(j)}((u, w))) \, || \, \varphi(c_t^{(j)}((v, w)))]) \, || \\ &\qquad\qquad f_2([\mathbf{B}_{u,w,:}^t \, || \, \mathbf{B}_{v,w,:}^t])] \, | \, w \in \mathcal{V}\}\}) \end{aligned} \tag{30}$$

Since the composition of injective functions is still injective, the above can be written as:

$$\boldsymbol{h}_t^{(j+1)}((u, v)) = f(c_t^{(j)}((u, v)), \{\{(c_t^{(j)}((u, w)), c_t^{(j)}((v, w)), \mathbf{A}_{u,w,:}^{<t}, \mathbf{A}_{u,w,:}^{<t}) \, | \, w \in \mathcal{V}\}\}) \tag{31}$$

Where $f$ is an injective function. The conversion from **B** to **A** is legal because of the bijective mapping from TIT to DAT (Eq. (6)). Therefore, we have

$$
\begin{aligned}
\boldsymbol{h}_t^{(j+1)}((u,v)) &= f \circ \mathrm{HASH}^{-1} \circ \mathrm{HASH}(c_t^{(j)}((u,v)), \\
&\qquad \{\{(c_t^{(j)}((u,w)), c_t^{(j)}((v,w)), \mathbf{A}_{u,w,:}^{<t}, \mathbf{A}_{u,w,:}^{<t}) \mid w \in \mathcal{V}\}\}) \\
&= f \circ \mathrm{HASH}^{-1}(c_t^{(j+1)}((u,v)))
\end{aligned}
\tag{32}
$$

By above equation, we have found an injective function $\varphi' = f \circ \mathrm{HASH}^{-1}$ for the $(j+1)$-th iteration, thus the proposition is proven. $\square$

## C  DETAILS OF EXPERIMENTAL SETTINGS

### C.1  DATASETS

The details of datasets included in our experiments will be introduced in the followings. The statistics of these datasets are summarized in Table 3.

**Reddit.**  Reddit[1] is a dataset of user activities that includes subreddits posted by various users within a single month on the Reddit website. It is a bipartite dataset that includes the 10,000 most active users and 984 subreddits, offering detailed interaction features.

**Wikipedia.**  Wikipedia[2] captures the clicking actions on Wikipedia pages by various users. It is a bipartite network which includes clicking actions on 1,000 pages over the course of one month with detailed interaction features provided by the users.

**UCI.**  UCI[3] is a non-bipartite network that encompasses sent messages between users within an online community of students from the University of California, Irvine. The nodes represent the students, and the edges denote the messages exchanged among them.

**Enron.**  Enron[4] is a non-bipartite collection comprising around 0.5M emails exchanged among employees of the Enron energy company over a period of three years.

**MOOC.**  MOOC[5] is a bipartite interaction network of online sources, where the nodes represent students and course content units. Each link indicates a student's access to a specific content unit and is characterized by a 4-dimensional feature.

**LastFM.**  LastFM[6] is a bipartite dataset that contains information about which songs were listened to by which users over the course of one month. In this dataset, users and songs are represented as nodes, and the links indicate the users' listening behaviors.

**CanParl.**  Canparl[7] is a dynamic political network documenting the interactions between Canadian Members of Parliament (MPs) from 2006 to 2019. In this network, each node represents an MP from an electoral district, and a link is formed when two MPs both vote "yes" on a bill. The weight of each link indicates the number of times one MP voted "yes" alongside another MP within a year.

---

[1] https://snap.stanford.edu/jodie/reddit.csv
[2] https://snap.stanford.edu/jodie/wikipedia.csv
[3] https://konect.cc/networks/opsahl-ucsocial/
[4] https://www.cs.cmu.edu/~enron/
[5] https://snap.stanford.edu/jodie/mooc.csv
[6] https://snap.stanford.edu/jodie/lastfm.csv
[7] https://github.com/shenyangHuang/LAD

Table 3: Statistics of the datasets. Average Interaction Intensity $\lambda = 2|\mathcal{E}|/(|\mathcal{V}|\mathcal{T})$, where $|\mathcal{E}|$ and $|\mathcal{V}|$ denote the number of interactions and nodes, respectively. $\mathcal{T}$ denotes the total duration (seconds).

| Datasets | #Nodes | #Links | #Node Feat. | #Edge Feat. | Bipartite | Duration | $\lambda$ |
|---|---|---|---|---|---|---|---|
| Wikipedia | 9,227 | 157,474 | 0 | 172 | True | 1 month | $4.57 \times 10^{-5}$ |
| Reddit | 10,984 | 672,447 | 0 | 172 | True | 1 month | $1.27 \times 10^{-5}$ |
| MOOC | 7,144 | 411,749 | 0 | 4 | True | 17 months | $2.62 \times 10^{-6}$ |
| LastFM | 1,980 | 1,293,103 | 0 | 0 | True | 1 month | $5.04 \times 10^{-4}$ |
| Enron | 184 | 125,235 | 0 | 0 | False | 3 years | $1.20 \times 10^{-5}$ |
| UCI | 1,899 | 59,835 | 0 | 0 | False | 196 days | $3.79 \times 10^{-6}$ |
| CanParl | 734 | 74,478 | 0 | 1 | False | 14 years | $4.59 \times 10^{-7}$ |

## C.2 IMPLEMENTATION DETAILS

**Baseline implementations.** We reproduce the experimental results of JODIE, DyRep, TGAT, TGN, CAWN, Graphmixer, TCL and DyGFormer based on the dynamic graph learning library DyGLib [8]. Specifically, we fix the optimizer, learning rate and batch size are set as 0.0001, 200 and Adam (Kingma & Ba, 2014), respectively, for all baselines. All the baseline methods for 50 epochs using the early stopping strategies of patience of 10. The model achieving the best performance on validation set is selected for testing. All other hyperparameters settings of the specific models, such as dimensions of various encodings or the number of sampled neighbors follow the optimal configurations provided by DyGLib. We repeat the experiments three times with different seeds. For PINT, we report the results in their paper except MOOC and CanParl. For results on MOOC and CanParl, we run official code of PINT [9]. All hyper-parameters are set to their default values.

**HopeDGN implementations.** We implement the HopeDGN based on DyGLib. The optimizer, learning rate, batch size, number of epochs and early stopping strategies are set same as baseline methods. The number of preserved timestamps $K$ in MITE is set as 32. The dimension of MITE $d_B$ is set as 50. The dimension of time encoding $d_T$ is set as 100. The aligned dimension $d$ is set as 50. The number of Transformer layer is 2. The number of attention heads is 2. The maximum input neighbor length $|\mathcal{N}|$ and the patching numbers $P$ of datasets are summarized in Table 4. All the experiments are conducted on a Linux Ubuntu 18.04 Server with a NVIDIA RTX2080Ti GPU.

Table 4: Maximum input neighbor length $|\mathcal{N}|$ and patching number $P$ of datasets..

| | Reddit | Wikipedia | UCI | Enron | LastFM | MOOC | CanParl |
|---|---|---|---|---|---|---|---|
| $|\mathcal{N}|$ | 64 | 32 | 32 | 256 | 128 | 256 | 2048 |
| $P$ | 2 | 1 | 1 | 8 | 4 | 8 | 64 |

# D ADDITIONAL EXPERIMENTS

## D.1 THE AUC RESULTS OF LINK PREDICTION.

The AUC results of link prediction experiments are presented in Table 5. We observe that the proposed HopeDGN achieves the best performance on all seven datasets for both transductive and inductive settings.

## D.2 NODE CLASSIFICATION

Following the settings of Rossi et al. (2020), we also compare the node classification performance of proposed HopeDGN and other baselines. In particular, the weights of DyGNN's encoder are pre-trained based on link prediction task. Then, a two-layer MLP is added on top of the encoder

---

[8] https://github.com/yule-BUAA/DyGLib
[9] https://github.com/AaltoPML/

Table 5: The AUC results of transductive/inductive link prediction are reported. The values are multiplied by 100. The results of the best and second best performing models are highlighted in **bold** and underlined, respectively.

|  | Model | Reddit | Wikipedia | UCI | LastFM | Enron | MOOC | CanParl |
|---|---|---|---|---|---|---|---|---|
| Transductive | JODIE | 98.20±0.05 | 95.36±0.12 | 90.15±0.07 | 71.95±1.82 | 84.68±2.97 | 82.80±0.83 | 78.15±0.21 |
| | DyRep | 98.02±0.15 | 93.60±0.09 | 65.19±4.26 | 71.06±0.21 | 81.87±1.76 | 83.30±0.82 | 78.19±3.07 |
| | TGAT | 98.13±0.02 | 96.46±0.15 | 78.30±0.48 | 71.25±0.26 | 66.13±1.66 | 85.37±0.48 | 75.61±3.87 |
| | TGN | 98.59±0.02 | 98.02±0.09 | 89.83±0.11 | 78.26±2.11 | 87.53±2.25 | 89.64±0.51 | 74.11±0.08 |
| | CAWN | 99.02±0.01 | 98.55±0.02 | 93.55±0.02 | 87.12±0.01 | 89.27±0.21 | 79.95±0.16 | 77.07±1.74 |
| | GraphMixer | 96.77±0.00 | 96.36±0.04 | 91.25±0.97 | 73.69±0.11 | 84.57±0.11 | 83.35±0.17 | 83.64±0.14 |
| | TCL | 96.88±0.02 | 95.53±0.23 | 84.18±0.39 | 70.24±2.21 | 73.92±0.28 | 82.54±0.07 | 73.43±0.93 |
| | DyGFormer | 99.15±0.01 | 98.84±0.00 | 93.98±0.20 | 91.38±0.03 | 93.08±0.05 | 85.71±0.43 | 97.71±0.29 |
| | HopeDGN | **99.27±0.00** | **99.17±0.02** | **96.51±0.08** | **92.92±0.01** | **93.59±0.04** | **90.93±0.25** | **98.52±0.59** |
| | Improve(%) | 0.12 | 0.33 | 2.53 | 1.54 | 0.51 | 1.29 | 0.81 |

|  | Model | Reddit | Wikipedia | UCI | LastFM | Enron | MOOC | CanParl |
|---|---|---|---|---|---|---|---|---|
| Inductive | JODIE | 96.40±0.13 | 92.97±0.21 | 77.14±0.95 | 81.31±0.89 | 81.00±2.55 | 83.64±1.03 | 53.64±2.34 |
| | DyRep | 95.85±0.31 | 90.76±0.05 | 56.23±1.18 | 82.41±0.08 | 72.54±3.60 | 82.78±1.72 | 55.16±1.63 |
| | TGAT | 96.52±0.13 | 95.77±0.09 | 77.16±0.15 | 76.64±0.22 | 59.32±0.23 | 84.91±0.71 | 55.59±1.17 |
| | TGN | 97.26±0.10 | 97.40±0.04 | 82.36±1.27 | 85.59±1.13 | 79.46±4.86 | 89.15±1.29 | 55.53±4.16 |
| | CAWN | 98.45±0.04 | 98.03±0.01 | 89.77±0.06 | 89.87±0.01 | 85.49±0.08 | 81.45±0.25 | 58.52±0.60 |
| | GraphMixer | 94.91±0.03 | 95.83±0.05 | 88.90±0.73 | 80.55±0.11 | 76.76±0.08 | 81.96±0.26 | 58.91±0.59 |
| | TCL | 93.86±0.38 | 95.51±0.12 | 80.35±0.71 | 76.27±1.98 | 70.38±0.55 | 80.93±0.04 | 56.10±0.05 |
| | DyGFormer | 98.68±0.00 | 98.42±0.02 | 91.49±0.19 | 92.95±0.03 | 90.32±0.28 | 85.60±0.43 | 88.99±0.14 |
| | HopeDGN | **98.88±0.01** | **98.82±0.00** | **94.12±0.03** | **94.33±0.01** | **90.51±0.15** | **91.11±0.36** | **89.16±0.73** |
| | Improve(%) | 0.20 | 0.40 | 2.63 | 1.38 | 0.19 | 1.96 | 0.17 |

for classification. Two datasets with node labels (Reddit and Wikipedia) are adopted for evaluation. Note that the representations obtained by HopeDGN are node-pair level, thus we make some modifications to incorporate the node classification experiments. Specifically, given a target node pair $(u, v)$ at time $t$, we modify Eq. (12) as:

$$
\begin{aligned}
\boldsymbol{h}_t(u) &= \text{MEAN}(\boldsymbol{H}^{(L)}_{1:|\mathcal{N}(u,t)|})\boldsymbol{W}_{out} + \boldsymbol{b}_{out} \\
\boldsymbol{h}_t(v) &= \text{MEAN}(\boldsymbol{H}^{(L)}_{|\mathcal{N}(u,t)|+1:|\mathcal{N}(v,t)|})\boldsymbol{W}_{out} + \boldsymbol{b}_{out}
\end{aligned}
\tag{33}
$$

to generate the representation of $(u, t)$ and $(v, t)$, respectively. The AUC results are presented in Table 6. We observe that HopeDGN achieves the highest average rankings compared to other baselines. In addition, The HopeDGN achieved the best performance on the Reddit and significantly outperformed other baselines.

Table 6: AUC results of node classification (multiplied by 100). The values of the best performing models are marked in **bold**. The average ranks are included.

|  | Wikipedia | Reddit | Avg. Rank |
|---|---|---|---|
| JODIE | **89.42±1.45** | 61.81±0.67 | 4.5 |
| DyRep | 85.66±1.55 | 65.73±1.88 | 4.5 |
| TGAT | 82.39±2.56 | 68.45±0.45 | 5.0 |
| TGN | 85.44±1.67 | 60.85±2.25 | 7.0 |
| CAWN | 83.57±0.22 | 66.22±1.04 | 5.5 |
| Graphmixer | 86.90±0.06 | 65.25±3.12 | 4.0 |
| TCL | 81.58±4.10 | 66.98±1.25 | 6.0 |
| DyGFormer | 85.29±2.79 | 64.51±2.89 | 6.5 |
| HopeDGN | 85.69±0.67 | **71.20±1.47** | **2.0** |

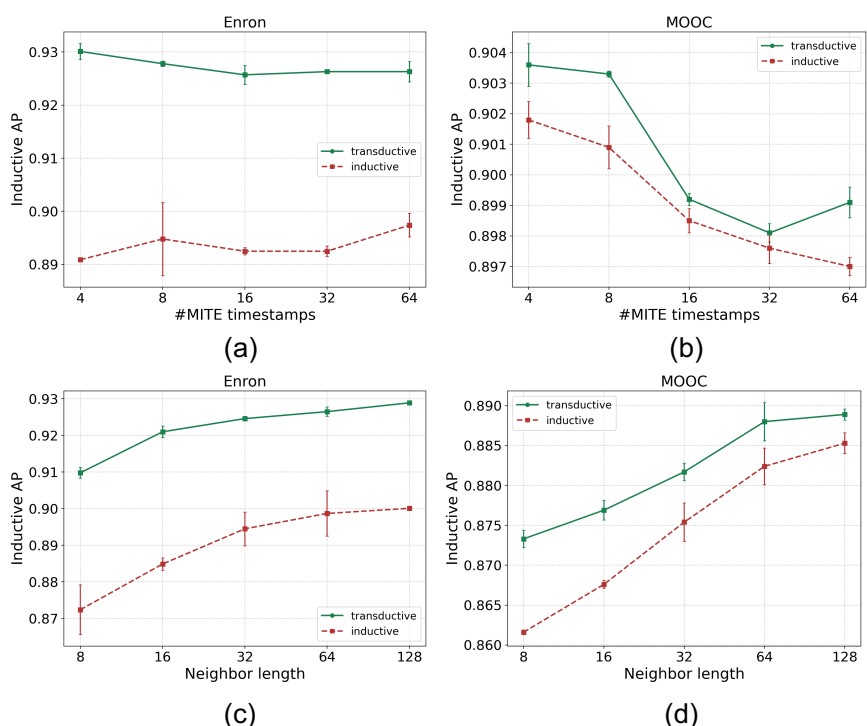

Figure 4: Parameter sensitivity of HopeDGN.

### D.3 PARAMETER SENSITIVITY

In this section, we evaluate the parameters sensitivity of HopeDGN, including non-infinite timestamps number in MITE $K$ and number of neighborhood, on Enron and MOOC datasets. $K$ is searched in range $\{4, 8, 16, 32, 64\}$ and the neighborhood length is searched in range $\{8, 16, 32, 64, 128\}$. The results are presented in Fig. 4. We observe that the performance of Hope-DGN is quite stable with varying $K$, on both Enron and MOOC dataset. Additionally, the performance of HopeDGN improves until converges when the input length of neighborhood increases, on both Enron and MOOC datasets. This is reasonable because a larger neighborhood receptive field can help the model more likely perceive non-isomorphic node pairs, thereby learning more expressive representations.

### D.4 EFFICIENCY EVALUATION

We compare the efficiency of proposed HopeDGN with other baselines. Specifically, we evaluate the training time per epochs (seconds) of different models on the MOOC dataset, and their inductive AP values are reported together. Note that the optimal parameter settings are adopted for baseline methods. The training time of HopeDGN with various input neighbor length ($\{16, 64, 128, 256\}$) are reported. The results are presented in Fig. 5. From Fig. 5 (a), we observe that HopeDGN with 256 neighbors is slightly faster than CAWN, but slower than other baselines. However, it can achieve the best performance among all baselines. Additionally, shortening the neighbor length of the HopeDGN can significantly reduce training time while still maintaining good performance. For example, The training time of the HopeDGN with 16 neighbors is only $\sim 24\%$ of the HopeDGN with 256 neighbors, significantly less than the CAWN and DyGFormer models, while its performance is significantly better than TGAT, DyGFormer and CAWN. From Fig. 5 (b), we observe that the training time of the HopeDGN approximately increases linearly with the neighbor length. This result is consistent with our complexity analysis in Sec. 4.4.

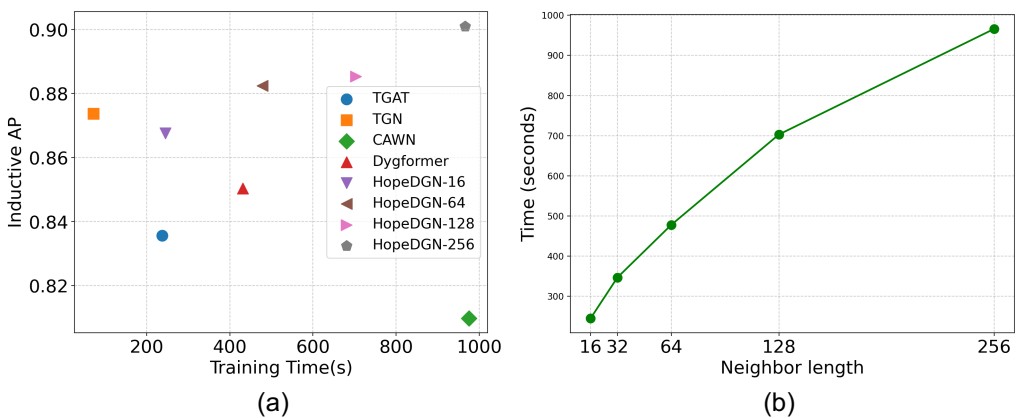

(a)              (b)

Figure 5: Left: Efficiency-performance comparison of different models on MOOC dataset. The X-axis is the training time per epoch (seconds). The Y-axis is the inductive AP value. 'HopeDGN-$n$' denotes HopeDGN with input neighbor length of $n$. Right: Training time of HopeDGN with various neighbor length on MOOC dataset.

