# OpenReview forum: "Towards Dynamic Graph Neural Networks with Provably High-Order Expressive Power"
_ICLR.cc/2025/Conference — ICLR 2025 Conference Withdrawn Submission_

### Official Review · Reviewer_f2fD · 2024-10-29

**Soundness:** 2
**Presentation:** 2
**Contribution:** 2
**Rating:** 3
**Confidence:** 5

**Summary:**

The authors present a new dynamic GNN model, HopeDGN, designed to exceed the expressive power limitations of current DyGNNs, which are capped by the 1-DWL test. HopeDGN aims to achieve higher-order expressive power by leveraging the 2-WL test.

**Strengths:**

1. This paper investigates an important problem.

2. The paper thoroughly explores the expressive power of dynamic GNNs in relation to the 2-WL test.

**Weaknesses:**

1. Similarity to Existing Models: The model structure closely resembles DyGFormer, particularly in Section 4.4, where essential component, from node encoding to the patching-based transformer encoder, are almost identical. However, the authors did not correctly cite DyGFormer, referencing only Dosovitskiy et al. (2021) instead.

2. Lack of Comprehensive 2-WL Analysis: While the paper’s primary contribution is introducing 2-WL testing for dynamic graphs, it falls short of a thorough examination of prior dynamic graph models’ failures under 2-WL. Some inaccuracies are noted; for example, in Figure 1, the claim that DyGNN cannot differentiate node pairs AC and AD is vague. With Neighbor Co-Occurrence Encoding, DyGFormer can indeed distinguish between these pairs. Additionally, the authors did not address scenarios where their model might also fail.

3. Limited Experiments: 3. According to the experimental setup in DyGFormer, there are three different negative sampling configurations. However, this paper appears to only conduct one configuration. Additionally, What’s the purpose of removing time encoding in the ablation study, as time encoding seems to be a common module in DyGNN frameworks and is not a contribution introduced by this paper.

4. Ambiguity in Notation: Certain symbols and terms are unclear, creating readability issues. For instance, the symbol S appears frequently in the complex analysis section (line 398) without definition. Similarly, i and j in Equation 3 lack clear explanations, and there are other instances of undefined notation throughout the paper.

5. Unclear Theoretical Framework: The authors state that they "establish a theoretical framework to quantify the expressive power of DyGNNs," but it is unclear how this quantification is achieved or where this analysis is elaborated upon in the paper.

**Questions:**

Please refer to the weakness

---

> ### Author Response · Authors · 2024-11-21
> **Response to Reviewer f2fD**
>
> > **W1: Similarity to Existing Models**.
>
> **Response**: We believe there exists misunderstandings. The proposed HopeDGN is **clearly different** from DyGFormer in following aspects. **Firstly**, HopeDGN is a general framework which can be implemented by many forms, while the model structure of DyGFormer is fixed. **Secondly**, HopeDGN adopts the proposed MITE as the correlation encoding while DyGFormer adopts Neighbor Co-Occurrence Encoding (NCOE). The advantage of MITE over NCOE is discussed in Section 4.2. **Thirdly**, the output of HopeDGN is the embeddings of a node pair, while the output of DyGFormer is node embedding.
>
> > **W2: Some inaccuracies in Fig.1**.
>
> **Response**:  Thanks for your comments. We apologize for making you confusion. Figure 1 illustrates the limited expressive power of **defined DyGNNs (Section 4.3)**, but not for DyGNNs with heuristic encodings such as DyGFormer. We will clarify this point in the revised paper. Although the heuristic encodings in DyGFormer have improve the expressive power to some extent, its expressive power is not quantifiable from a theoretical perspective. We will add DyGFormer for discussion in the revised paper.
>
> > **W3: Limited Experiments**
>
> **Response**: Thanks for your suggestion. We will consider to add the experiments of different negative sampling configurations in the revised paper.
>
> > **W4: Ambiguity in Notation**
>
> **Response**:  Thanks for your suggestion. We will revise these symbols in the revised paper.
>
> > **W5: Unclear Theoretical Framework**
>
> **Response**: Thanks for your suggestion. To quantify the expressive power of DyGNNs, we propose the k-DWL for dynamic graphs, which is an extension of WL test to dynamic graphs. We have demonstrated in Proposition 1 that the expressive power of (k+1)-DWL is strictly stronger than k-DWL. This demonstrate that the proposed DWL test is a valid quantification of expressive power, similar to WL test in static graph used to quantify the expressive power of static graphs.

---

### Official Review · Reviewer_1NPK · 2024-10-30

**Soundness:** 3
**Presentation:** 2
**Contribution:** 2
**Rating:** 5
**Confidence:** 4

**Summary:**

This work aims to address the gap in dynamic graph learning research concerning the theory of expressive power. The authors first introduce the $k$-DWL test and argue that the expressive power of existing DGNN models is bounded by the 1-DWL test. To advance this, they propose a new DGNN model that works analogously to the 2-DWL test and prove that the proposed model, Global HopeDGN, is equivalent to the 2-DWL test. The experiments show the local variant of HepeDGN can achieve superior performance across several datasets.

**Strengths:**

1. This work is the first attempt at assessing the expressive power of dynamic graph learning and introduces the concept of the $k$-WL test. It intuitively points out the limitations of existing DGNN models and proves that the 1-WL test limits their expressive power.
2. This work proposes a new DGNN model, HopeDGN, inspired by the 2-WL test. This model has greater expressive power than vanilla DGNN models, and the experiments show its superiority.

**Weaknesses:**

1. Although no research exists on the expressive power theory of DGNN, a few studies have considered the correlations between historical neighbors of two terminal nodes, such as DyGFormer, HOT [R1], and CAWN. HopeDGN is somewhat of an incremental study compared to DyGFormer. HopeDGN exhibits higher performance in the experiments but may incur much larger training costs.
2. The discussions of the expressive power of existing DGNN models are not completely correct.
   1. The abstraction of DyGNNs in Section 3 cannot cover existing studies. For instance, CAWN can leverage the information from high-order neighbors in a single layer if the length of the random walk is larger than 1.
   2. DyGFormer, HOT, and CAWN are able to distinguish AC and AD in Figure 1.
3. Some notations are confusing. $l$ is used to represent the layer of AGG and UPDATE in Line 167 and the node labeling function in Line 239. And the layer of AGG and UPDATE of local HopeDGN in the experiments is not specified. In addition, the MITE of $w$ is denoted as $X_{B,w}$ in Line 369 and $X_{M,w}$ in Line 289.

[R1] HOT: Higher-Order Dynamic Graph Representation Learning with Efficient Transformers

**Questions:**

1. The expressive power of the 1-WL test and the 2-WL test is the same in static graphs, as argued in [R2]. Is the 1-DWL test equivalent to the 2-DWL test? Intuitively, it is.
2. Could the authors provide the training cost of HopeDGN and the baselines?


[R2] A Short Tutorial on The Weisfeiler-Lehman Test And Its Variants.

---

> ### Author Response · Authors · 2024-11-21
> **Response to Reviewer 1NPK**
>
> > **W1: HopeDGN is somewhat of an incremental study compared to DyGFormer. HopeDGN exhibits higher performance in the experiments but may incur much larger training costs.**
>
> **Response**: We believe there exists misunderstandings. Here we clarify the difference between the proposed HopeDGN and other baselines (DyGFormer, HOT and CAWN) as follows. Firstly, the expressive power of HopeDGN is **proven to be equivalent to 2-DWL test**, while other baselines (DyGFormer, HOT, CAWN) only leverage heuristic encodings **without theoretical derivations**. Secondly, the proposed HopeDGN **is a general framework which can be implemented by many forms**. The specific implementation of HopeDGN in section 4.4 is just one form of it. Thus, HopeDGN is not an incremental study to DyGFormer.
>
> In terms of training costs, we have evaluated the training efficiency of HopeDGN in experiments and the results are presented in Figure 5. In Figure 5, we observe that HopeDGN with neighbor length 16 is faster than DyGFormer.
>
> > **W2: The discussions of the expressive power of existing DGNN models are not completely correct.**
>
> **Response**: Thanks for your suggestion. Our abstraction of DyGNNs in Section 3 closely follows existing works [1]. Our discussion about the expressive power (including Fig 1 and Proposition 1) is for this definition of DyGNN, but not for DyGNNs with heuristic encodings (such as DyGFormer, HOT, CAWN). We will clarify this point in the revised paper.
>
> > **W3: Some notations are confusing**.
>
> **Response**: Thanks for your suggestion. We will specify these notations in the revised paper.
>
> > **Q1: Is the expressive power of 1-DWL test equivalent to the 2-DWL test?**.
>
> **Response**:   Thanks for your question. The answer is no. **Our proposed 2-DWL test is an extension of Folklore variant of 2-WL test**, whose expressive power is proven stronger than 1-WL test [2].
>
> > **Q2: Could the authors provide the training cost of HopeDGN and the baselines?**.
>
> **Response**:   Thanks for your question. We have provided the training cost in Fig.5 of Appendix.
>
> **Ref**:
> [1] Souza A, Mesquita D, Kaski S, et al. Provably expressive temporal graph networks[J]. Advances in neural information processing systems, 2022, 35: 32257-32269.
> [2] Morris C, Lipman Y, Maron H, et al. Weisfeiler and leman go machine learning: The story so far[J]. The Journal of Machine Learning Research, 2023, 24(1): 15865-15923.

---

### Official Review · Reviewer_Aj2F · 2024-10-31

**Soundness:** 3
**Presentation:** 3
**Contribution:** 3
**Rating:** 6
**Confidence:** 4

**Summary:**

The paper introduces a novel Dynamic Graph Neural Network (DyGNN) framework called HopeDGN (Dynamic Graph Neural Network with High-order Expressive power). The primary focus of the paper is to address the limited expressive power of existing DyGNNs in capturing evolving patterns in dynamic graphs. The authors propose the k-dimensional Dynamic WL tests (k-DWL) as a theoretical framework to quantify the expressive power of DyGNNs, and introduce the Multi-Interacted Time Encoding (MITE) that captures the bi-interaction history of target node pairs with other nodes. MITE is integrated into the HopeDGN framework, and theoretical results show that HopeDGN can achieve expressive power equivalent to the 2-DWL test. Experimental results demonstrate that HopeDGN achieves superior performance on link prediction and node classification tasks compared to other baselines.

**Strengths:**

- The Multi-Interacted Time Encoding (MITE) allows the model to capture indirect dependencies between node pairs, which is crucial for tasks like link prediction. This module is a plug-and-play component that can be integrated into various models, enhancing their expressive power.

- The paper provides proofs that HopeDGN can achieve expressive power equivalent to the 2-DWL test, which is a significant improvement over existing DyGNNs. This theoretical grounding adds credibility to the practical results.

- The authors discuss multiple model design details including Neighborhood encoding, patching and Transformer encoder. The authors conduct extensive experiments on both link prediction and node classification tasks across multiple datasets. The results consistently show that HopeDGN outperforms existing baselines, demonstrating its effectiveness.

**Weaknesses:**

- The paper could benefit from a detailed comparison with other high-order GNNs that have been proposed for static graphs since the basic idea of encoding node pairs comes from high-order WL tests, like https://arxiv.org/abs/1810.02244.
- See questions below.

**Questions:**

- I am confused with the example in Figure 1. Suppose the graph is static, according to symmetry 1-WL test will give identical labels to C and D, then GNN cannot distinguish node pairs (A, C) and (A, D) using only node embeddings, which has no connection to graph dynamics. And since 2-WL works on node pairs, we can easily verify that (A, C) and (A, D) will get different labels in 2-WL test so methods derived from 2-WL can distinguish them (like MITE). Therefore I don’t think that this example can well explain the expressive limitations of 1-DWL since it also works in static graphs. Can the authors provide more explanations?

- Can the proposed method work on dynamic graphs in which edge interaction can both start and end?

---

> ### Author Response · Authors · 2024-11-21
> **Response to Reviewer Aj2F**
>
> > **W1: The paper could benefit from a detailed comparison with other high-order GNNs for static graphs**.
>
> **Response**: Thanks for your suggestion. We will compare some papers you mentioned in the revised paper in detail.
>
> > **W2: Confusion about Fig.1**.
>
> **Response**: Thanks for your suggestion. The proposed DWL test is inspired by the WL test in static graphs, which additionally hashes the timestamps arrays. We acknowledge that the failure case of 1-DWL test may be somewhat similar to 1-WL test. However, the dynamic graph has clear difference with static graphs since each interaction (edge) is associated with timestamps array. Since **the focus of this study is on dynamic graphs**, and Figure 1 indeed presents a failure case of vanilla DyGNN (or 1-DWL test), we think this is appropriate for illustrating the limited expressive power of DyGNN, regardless whether it can be work on static graphs.
>
> > **W3: Can the proposed method work on dynamic graphs in which edge interaction can both start and end?**
>
> **Response**: Thanks for your suggestion. To address the edge deletion event, we need to make some modification on the current model. In specific, we can add an 0-1 indicator on mite to signify whether each event is an edge addition or deletion event. Through this technique, the model can capture whether the event is an addition or deletion event.

---

### Official Review · Reviewer_W1oK · 2024-11-02

**Soundness:** 3
**Presentation:** 2
**Contribution:** 2
**Rating:** 5
**Confidence:** 3

**Summary:**

The paper presents a novel dynamic graph neural network that is theoretically inspired by a proposed dynamic-WL test. According to this framework, they show that existing DyGNNs are bounded by 1-DWL test and by using a multi-interaction time encoding (MITE), they can increase the expressiveness to up to 2-DWL test. Then, they propose HopeDGN, which updates the representation of a node pair by aggregating not just the historical interactions but also the elapsed times. Through a transformer-based implementation, they show improved performance across seven datasets, while demonstrating plug-and-play benefits.

**Strengths:**

- The proposed MITE matrix can be used in a plug-and-play fashion to provide gains in existing architectures.
- Time complexity and efficiency analysis are provided.
- HopeDGN leverages the strengths of the transformer and a theoretically motivated WL-test to propose a more effective method.
- Empirical comparison is thorough in both transductive and inductive settings of link prediction against representative baselines.

**Weaknesses:**

- The DWL test is almost identical to the temporal WL test in PINT [1]. This is not acknowledged in the main paper and thus, can be flagged as plagiarism.
- Discussion with existing related work is casual and not carefully positioned.
  - It is mentioned that it is not clear how relative positional features provide theoretical benefits even though it was shown theoretically in Souza et al., 2022 [1].
  - It is mentioned that "Souza et al. (2022) proves that adding a memory mechanism will not change the expressive power of DyGNNs. Therefore, the expressive power of DyGNNs can be fully characterized by the 1-DWL test." but this is wrong as they show that is true specifically when the architecture of the MP-GNN is deep enough.
  - It is not clear whether what are the theoretical benefits of the proposed method as compared to [1] or [4]. It seems that the benefits are complementary but needs further discussion. Does the proposed method already provably reach the expressiveness of time-then-graph and PINT in their expressiveness tests.
- Space complexity is not provided nor computationally compared. The additional cost of storing B matrix along with the transformer architectures may be prohibitive in many cases and limit the scalability of the proposed method.
- MITE encoding is similar to the time projection for staleness as done in JODIE [2] and TGN [3]. However, these similarities are neither acknowledged nor discussed in the paper. It seems like MITE is an extension of the time projection to the neighbors' encodings in addition to only the interacting node's embeddings.
- The proof of the main proposition 3 that shows the theoretical disadvantage of existing dyGNNs seems wrong. In particular, they only consider node b for graph (a) and node h in graph (b). On the other hand, Equation 9 denotes that they should have considered all historical neighbors w for both graphs. This would have meant considering nodes b, h, d for graph (a) and nodes b, h, f for graph (b). In that case, d and f seem to be the difference maker and not b and h.
- Considering the MITE interactions for all historical neighbors seem unscalable as the size of the graph increases.
- Running time of HopeDGN is not compared against the baselines, only the training time is compared.

[1] Souza, Amauri, et al. "Provably expressive temporal graph networks." Advances in neural information processing systems 35 (2022): 32257-32269.

[2] Kumar, Srijan, Xikun Zhang, and Jure Leskovec. "Predicting dynamic embedding trajectory in temporal interaction networks." Proceedings of the 25th ACM SIGKDD international conference on knowledge discovery & data mining. 2019.

[3] Rossi, Emanuele, et al. "Temporal graph networks for deep learning on dynamic graphs." arXiv preprint arXiv:2006.10637 (2020).

[4] Gao, Jianfei, and Bruno Ribeiro. "On the equivalence between temporal and static graph representations for observational predictions." arXiv preprint arXiv:2103.07016 (2021).

**Questions:**

see above weaknesses.

---

> ### Author Response · Authors · 2024-11-21
> **Response to Reviewer W1oK (Part 1/2)**
>
> > **W1: The DWL test is almost identical to the temporal WL test in PINT [1]. This is not acknowledged in the main paper and thus, can be flagged as plagiarism.**
>
> **Response**: We believe there exists serious misunderstandings. There are some **clear difference** between temporal WL and PINT. **Firstly**, the proposed DWL test is $k$-order which colors the $k$-node set on dynamic graphs, while temporal WL test is only applicable for coloring single node. **Secondly**, temporal WL test hashes the single time interval while DWL test hashes the complete interaction timestamps array (a vector). Therefore, we think the proposed DWL should definitely **not** be recognized as plagiarism.
>
> > **W2: Discussion with existing related work is casual and not carefully positioned.**
>
> **Response**: We believe there exists misunderstandings. We respectively respond to your concerns as follows.
>
> + **Our claim in Paragraph 2 of Introduction is “it remains unclear how the relative positional features (RPF) quantitatively affect DyGNNs' expressive power” but not your paraphrase.** Our original claim is true because the Proposition 9 of PINT only shows that combining RPF with DyGNN is strictly more expressive than DyGNNs, but the exact expressive power of DyGNN with RPF is not quantified by some benchmarking criterion such as DWL test.
>
> + As you have stated, the expressive power of “sufficiently deep DyGNN” will not be be improved with memory. In addition, the expressive power of “sufficiently deep DyGNN” is bounded by 1-DWL test as shown in proposition 1. This means that the expressive power of “sufficiently deep DyGNN with memory” is bounded by 1-DWL test. Thus, we can easily conclude that the expressive power of “normal DyGNN with memory” is bounded by 1-DWL test. Thus, our claim is true.
>
> + **The goal of our work is not to prove that HopeDGN is more expressive than [1] and [4].** Instead, our goal is to propose a model (HopeDGN) whose high-order expressive power can be quantified by benchmarking criterion such as DWL test, while both [1] and [4] cannot address.
>
> > **W3: The additional cost of storing B matrix may be prohibitive.**
>
> **Response**: In implementations, we do not use the original $B$ tensor in Eq. (6) but cut down the last $K$ non-infinite timestamps when computing MITE (as we have stated in Section 4.2). The MITE is computed in the current batch of data and discarded when moving into the next batch, thus the memory burden is light.

---

> ### Author Response · Authors · 2024-11-21
> **Response to Reviewer W1oK (Part 2/2)**
>
> > **W4: MITE encoding is similar to the time projection for staleness as done in JODIE [2] and TGN [3].**
>
> **Response**: We believe there exists misunderstandings. **MITE is completely different from time projection in JODIE and TGN.** Specifically, time projection aims to project the time interval between the last interaction and current time for central node (a single scalar) to make prediction, while MITE encodes the complete history between the neighbors to central node pair (a matrix). The computation procedures of these two techniques are completely different, and **there is no evidence that our MITE is the extension of time projection.**
>
> > **W5: The proof of the main proposition 3 that shows the theoretical disadvantage of existing dyGNNs seems wrong. In particular, they only consider node b for graph (a) and node h in graph (b). On the other hand, ...**
>
> **Response**: We believe there exists misunderstandings. The proposition 3 only aims to show that **there exists** an non-isomorphic node pair that DyGNN with MITE can distinguish while vanilla DyGNN cannot. We prove this by showing the case of (a,c) in Fig.3 (a) and (a,g) in Fig.3 (b). For your comment of “on the other hand …”, we are discussing DyGNN and DyGNN with MITE in Proposition 3, and HopeDGN (Eq. 9) is irrelevant to Proposition 3.

---

> > ### Comment · Reviewer_W1oK · 2024-11-25
> >
> > I thank the authors for their rebuttal despite their declared withdrawal above. Since they have already withdrawn the submission, I am unsure if they want to engage in a discussion but I would like to respond to their comments in case they will be helpful for future revisions.
> >
> > > Clear difference between PINT temporal WL and dynamic WL
> >
> > I don't agree there is a *clear* and obvious difference that should not be highlighted. The k-order variant is different but the 1-order variant is exactly the same as can be noted below:
> >
> > Proposed (1): $c_{t}^{(j)}(u) = \text{HASH} (c_{t}^{(j-1)}(u), \{(c_{t}^{(j-1)}(w), A_{u, v, :}^{<t}) | (v, \cdot) \in N(u, t)\})$
> >
> > PINT (2): $c_{t}^{(j)}(u) = \text{HASH} (c_{t}^{(j-1)}(u), \{(c_{t}^{(j-1)}(w), e_{u, v}(t'), t') | (u, v, t') \in G(t)\})$
> >
> > But $A_{u, v, :}^{<t}$ consist of all timesteps that $u$ and $v$ have interacted at so while (1) would hash a list of $(c_{t}^{(j-1)}(w), t_1, t_2, \cdots)$, (2) would hash $[(c_{t}^{(j-1)}(w), t_1), (c_{t}^{(j-1)}(w), t_2)]$. But to the hash function, these should not be different since if two graphs are found to be identical by HASH (1) then they would also be identical by HASH (2) and vice-versa. If this is not the case and the authors still want to claim the novelty of their 1-WL, then they should prove it.
> >
> > > Discusssion
> >
> > I understand now that the aim is to assign a number to the theoretical benefits and the claim can be made true but I still feel this can be improved for more clarity. Secondly, the Proposition 1 of your work cannot be used to discuss previous work.
> >
> > > Only storing K interactions
> >
> > This is an important point that should be discussed more, specifically studying the sensitivity of the results to this hyperparameter and how is the trade-off.
> >
> > > MITE as an extension of projection
> >
> > I still believe there are similarities and these are actually highlighted in the rebutted response. I am not saying it reduces the novelty of the current work but rather discussing that there have been earlier works that have projected past interactions of the node would enhance the completeness of the paper

---

### Official Review · Reviewer_Hh9m · 2024-11-04

**Soundness:** 3
**Presentation:** 3
**Contribution:** 2
**Rating:** 5
**Confidence:** 3

**Summary:**

This paper introduces a k-dimensional dynamic Weisfeiler-Lehman (WL) test as a novel approach to quantify the expressiveness of dynamic graph neural networks (DyGNNs) and unifies simpler DyGNNs under the 1-Dimensional WL (1-DWL) framework. Additionally, the paper proposes HopeDGN, which employs a more granular encoding method to achieve the expressiveness of a 2-DWL test. Experimental results indicate the method’s effectiveness.

**Strengths:**

1. The paper is the first to propose a k-dimensional dynamic WL test to evaluate the expressiveness of DyGNNs, addressing a relatively underexplored area in DyGNN expressiveness.
2. The proposed framework theoretically attains 2-DWL expressiveness by introducing a novel encoding scheme, which also shows potential for generalization to other models.
3. The model includes an optimized local version for practical applications, demonstrating strong performance.

**Weaknesses:**

1. While prior DyGNNs may lack a unified framework, they have implemented numerous techniques to enhance expressiveness. Examples include DyGformer [1] with neighbor co-occurrence encoding, CAWN [2] with anonymous walk paths, and NAT [3] with neighborhood-aware encoding, etc. These methods are not addressed within the proposed framework, potentially making it appear somewhat isolated.
2. The theoretical time complexity of the k-DWL framework, especially over longer time spans, may be a concern, as noted by the authors. While local variants are proposed, practical usability could still be limited by these computational demands.

[1] Towards Better Dynamic Graph Learning: New Architecture and Unified Library https://arxiv.org/abs/2303.13047
[2] Inductive Representation Learning in Temporal Networks via Causal Anonymous Walks https://arxiv.org/abs/2101.05974
[3] Neighborhood-aware Scalable Temporal Network Representation Learning https://arxiv.org/abs/2209.01084

**Questions:**

1. In Table 2, integrating MITE with other baselines shows significant improvements. How is this integration implemented, and does MITE demonstrate clear benefits compared to other encoding approaches?
2. Figure 5 highlights a notable efficiency improvement when reducing the neighbor length. What are the corresponding performance metrics associated with these efficiency gains?

---

> ### Author Response · Authors · 2024-11-21
> **Response to Reviewer Hh9m**
>
> > **W1: Some baselines are not addressed.**
>
> **Response**: Thanks for your comments. We acknowledge that some heuristic encodings have been designed to somewhat improve the expressive power of DyGNNs, such as those in CAWN, NAT, DyGFormer and PINT. **However, the expressive power of these models is not quantified (such as k-WL test in static graph).**
>
> To bridge this research gap, we propose a benchmarking hierarchy named **k-DWL test** to quantify the level of expressive power of DyGNNs, which has been highlighted in the second paragraph of introduction. Then, **we propose HopeDGN which is proven to be as expressive as 2-DWL test, thus its expressive power can be clearly quantified.**
>
> We will revise the introduction section and include these works for discussion.
>
> > **W2: The time complexity of HopeDGN.**
>
> **Response**: We argue that the time complexity of HopeDGN is controllable when the time spans is large. As analyzed in Section 4.4, the time complexity of HopeDGN is **linear with $S$** where $S$ is the length of concatenated historical neighbors. **Fig.5 also demonstrate that training time is linear with $S$ by experiments.** This linear complexity is similar to other DyGNNs such as DyGFormer and be scalable to the case of long time span.
>
> > **Q1: How is the integration of MITE implemented, and does MITE demonstrate clear benefits compared to other encoding approaches?**
>
> **Response**: The integration process is as follows: Given a DyGNN model A, suppose the target link to be predicted is $(u,v,t)$, where u and v are nodes and t is the timestamp. A processes by sampling the historical neighbors of $u$ and $v$, denoted as $\mathcal{N}(u,t)$ and $\mathcal{N}(v,t)$ respectively, For any $w \in \mathcal{N}(v,t) \cup \mathcal{N}(v,t)$, we can compute its MITE with respect to $(u,v)$ at time $t$ (eq. 7). Then, MITE is integrated with the node feature.
>
> To demonstrate the benefits of MITE over other encodings, we will also integrate Neighbor CoOccurrence Encoding (NCOE) with various baselines on various datasets in the revised paper.
>
> > **Q2: The performance metrics in Fig 5.**
>
> **Response**:  The inductive AP metrics of CorDGT with various neighbor length on MOOC (in Fig. 5) is summarized as follows.
>
> |              | HopeDGN-16 | HopeDGN-64 | HopeDGN-128 | HopeDGN-256 |
> |--------------|------------|------------|-------------|-------------|
> | Inductive AP | 86.76      | 88.24      | 88.53       | 90.10       |.

---

### Author Response · Authors · 2024-11-21
**General Response (Part 1/2)**

We thank all reviewers for taking the time to review our paper and for providing valuable and insightful feedback. We recognize that the quality of this paper can be further improved, and we decide to withdraw this paper.

Below are some overall clarifications for our work.

**Motivation**.

+ The expressive power of vanilla message-passing based DyGNNs is limited. Recently, some heuristic encodings have been designed to somewhat improve the expressive power of DyGNNs, such as DyGFormer [1], CAWN [2], PINT [3], **but the expressive power of these models is not quantified by some benchmarking criterions (such as k-WL tests in static graph).**

**Contribution of this work**.

+ In analogy to WL tests in static graph, we propose a benchmarking hierarchy named **k-DWL test** to quantify the level of expressive power of DyGNNs. Then, we propose **HopeDGN** which is proven to be as expressive as 2-DWL test, thus its high-order expressive power can be clearly quantified.

**Ref:**
+ [1] Yu L, Sun L, Du B, et al. Towards better dynamic graph learning: New architecture and unified library[J]. Advances in Neural Information Processing Systems, 2023, 36: 67686-67700.
+ [2] Wang Y, Chang Y Y, Liu Y, et al. Inductive representation learning in temporal networks via causal anonymous walks[J]. arXiv preprint arXiv:2101.05974, 2021.
+ [3] Souza A, Mesquita D, Kaski S, et al. Provably expressive temporal graph networks[J]. Advances in neural information processing systems, 2022, 35: 32257-32269.

---

> ### Author Response · Authors · 2024-11-21
> **General Response (Part 2/2)**
>
> Although we have decided to withdraw this paper, we still provide detailed responses below to address the concerns of reviewers.  We hope these clarifications and responses will help the reviewers and future readers to better understand this paper.
>
> Again, we thank the reviewers for providing valuable comments.

---

### Note · Authors · 2024-11-25

I have read and agree with the venue's withdrawal policy on behalf of myself and my co-authors.